# In-Context Learning as Rate–Distortion Optimization

**Jiayu Zhang** [* 1]  **Changbang Li** [* 2]  **Canran Xiao** [† 3]

## Abstract

In-context learning (ICL) is a practical way to adapt large models, yet under strict context limits, it remains unclear how to spend scarce tokens without being misled by noisy, redundant, or conflicting demonstrations. We address this gap by targeting *token-budgeted* context construction: how to select and compress demonstrations so the prompt carries maximal task-relevant signal with minimal predictive distortion. We propose RDCO, a deterministic, training-free optimizer that scores demonstrations by marginal task information per token, penalizes redundancy and prefix-conditioned conflicts, and finally compacts the selected context under a bounded predictive-divergence constraint to control drift. Across a 10-dataset ICL suite spanning classification and structured generation, RDCO achieves the best average performance (63.26 Acc. on classification and 60.26 EM on generation), improving over the strongest classification and generation baselines by 2.20 and 2.26 points, respectively, and improving the 10-task overall average by 4.94 points under the same budget. Our results suggest that viewing prompts as finite-capacity messages yields a principled and effective path to more reliable and token-efficient ICL.

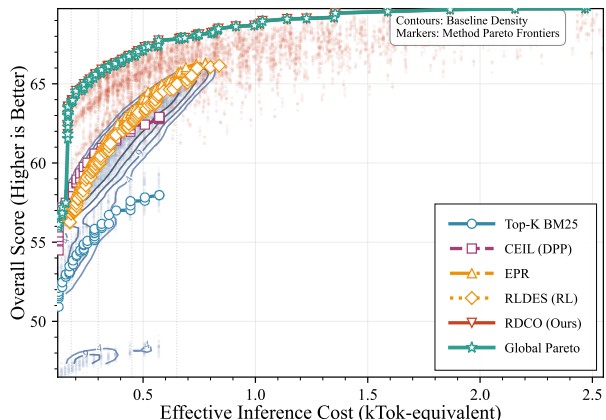

*Figure 1.* **Efficiency frontier.** RDCO achieves higher overall performance at the same inference cost, tracing a better Pareto frontier than strong baselines.

## 1. Introduction

In-context learning (ICL) has become a standard interface for adapting large (multi)modal models to new tasks without updating parameters, making it attractive for rapid deployment, privacy-sensitive settings, and retrieval-augmented applications where supervision is scarce or drifting (Dong et al., 2024; Wies et al., 2023). Yet modern ICL pipelines

increasingly operate under strict context limits and latency budgets, forcing systems to choose which demonstrations to include and how to fit them into a finite number of tokens. As a result, performance is often determined less by model capacity than by how efficiently the prompt encodes task-relevant evidence under a hard token budget (Yao et al., 2024; Cobbina & Zhou, 2025; Tan et al., 2025; Li et al., 2026b).

Recent work has made substantial progress on learning and retrieving better demonstrations, including supervised prompt retrievers and rerankers (Rubin et al., 2022; Wang et al., 2024; 2025; Yin et al., 2025) as well as set-level selection objectives that encourage diversity (Ye et al., 2023). Parallel lines of research aim to improve robustness to distribution shift (Byun et al., 2026) and stabilize predictions via calibration in dynamic prompting settings (Zhao et al., 2021; Zhou et al., 2024; Abbas et al., 2024; Tan et al., 2025). However, these advances leave a central tension unresolved: in practice, the context is a *finite-capacity message*, and adding tokens can help only when they contribute new task signal, while redundant or conflicting demonstrations can distort the implicit task hypothesis and waste scarce budget. Most existing selectors and calibrators do not explicitly model this *token-level* trade-off, and are therefore brittle when the candidate pool is noisy, near-duplicate, or partially mismatched. Moreover, prompt compression methods can

---

[1]Peking University, Beijing, China [2]University of Pennsylvania, Philadelphia, PA, USA [3]Shenzhen Campus of Sun Yat-sen University, Shenzhen, Guangdong, China. *Equal contribution. †Correspondence to: Canran Xiao <xiaocr3@mail.sysu.edu.cn>.

*Proceedings of the $43^{rd}$ International Conference on Machine Learning*, Seoul, South Korea. PMLR 306, 2026. Copyright 2026 by the author(s).

reduce inference cost (Jiang et al., 2023; Mu et al., 2023; Zang et al., 2025), but they are typically decoupled from per-query task evidence and may introduce uncontrolled prediction drift. Finally, evaluation studies show that prompt construction details (format, placement, and dynamic contexts) can dominate reported gains (Cobbina & Zhou, 2025), underscoring the need for a principled framework that treats context construction as an optimization problem under strict budgets rather than an ad-hoc engineering choice.

> **This paper asks:**
>
> *How should an ICL system allocate a fixed token budget to convey maximal task-relevant information while minimizing distortion from redundancy, conflicts, and compression-induced drift?*

We address this question by reframing token-budgeted context construction through a rate–distortion lens, and designing a deterministic, training-free optimizer that builds a compact context by prioritizing evidence that most improves query predictions per token while discouraging redundant and inconsistent signals and controlling drift under compaction. As shown in Fig. 1, our method achieve a better Pareto frontier than strong baselines.

Our contributions are as follows: (i) We cast ICL under strict context limits as a finite-capacity communication problem, clarifying why additional tokens can both help and hurt, and motivating token-aware context optimization rather than fixed-shot heuristics. (ii) We introduce a training-free, deterministic context construction pipeline that jointly supports budgeted demonstration selection and controlled compaction, targeting stable prompt behavior under noisy and dynamically assembled candidate pools. (iii) Across diverse ICL suites spanning classification, structured generation, calibration, and OOD retrieval settings, our approach improves token efficiency and achieves stronger performance–cost trade-offs compared to competitive selection and calibration baselines (Rubin et al., 2022; Ye et al., 2023; Wang et al., 2024; 2025; Tan et al., 2025).

## 2. Related Work

**Demonstration retrieval and set construction under finite context.** Early work largely relied on heuristic similarity retrieval (lexical or embedding-based), but performance was soon found to depend sharply on the *chosen set* of demonstrations rather than retrieval alone. This motivated learning-based prompt retrievers that use LLM feedback to train dense retrieval models, such as EPR (Rubin et al., 2022), LLM-R (Wang et al., 2024), and multi-task retrievers like UDR (Li et al., 2023) that improve transfer and deployment efficiency. In parallel, set-level objectives were

introduced to balance relevance and diversity, including DPP-based selection (e.g., CEIL (Ye et al., 2023) and two-stage DPP selection (Yang et al., 2023)), iterative selection strategies that adapt the diversity–similarity axis to each query (Qin et al., 2024), and more recent decision-theoretic formulations such as RL-based selection (Wang et al., 2025). For robustness beyond IID retrieval, CCL (Byun et al., 2026) learns causal representations to select demonstrations that generalize better under distribution shift, while mechanistic analyses such as Liu et al. (2025); Li et al. (2024); Zang (2025) distill what learning-based retrievers implicitly capture (e.g., task-agnostic vs. task-specific similarity signals). Despite this progress, most selection methods either require retriever training and labeled signals, or optimize relevance/diversity under a *fixed-shot* abstraction, leaving token cost, redundancy saturation, and *prefix-conditioned conflicts* largely implicit. Our work addresses this gap by formulating token-budgeted context construction as a rate–distortion trade-off and using a deterministic, training-free optimizer that scores *marginal information per token*, explicitly penalizes redundancy and conflicts, and selects under a strict budget.

**ICL brittleness, calibration, and prompt compression.** A complementary thread studies why ICL is brittle to prompt design and how to stabilize predictions. Contextual calibration (Zhao et al., 2021) and Batch Calibration (Zhou et al., 2024) mitigate prompt-induced priors, while LinC (Abbas et al., 2024) and Surprise Calibration (Tan et al., 2025) further improve reliability in dynamic ICL settings with changing contexts. Recent evaluation work shows that prompt format and demo placement can dominate reported gains, motivating more careful and consistent protocols (Cobbina & Zhou, 2025; Li et al., 2026a). Separately, prompt compression methods (e.g., LLMLingua (Jiang et al., 2023) and learned gist tokens (Mu et al., 2023)) reduce inference cost in long-context scenarios, but are typically not coupled to per-query task signals and may induce uncontrolled output drift. In contrast, our approach integrates selection and compaction: we compact the selected context with an explicit predictive-divergence constraint, enabling token savings while controlling distributional drift in the model's query prediction.

## 3. Method

We formulate in-context learning (ICL) under a strict token budget as a rate–distortion trade-off: the context is a finite-capacity message that should convey task-relevant information while minimizing predictive distortion induced by noisy, redundant, or conflicting demonstrations. We propose a deterministic context optimizer that (i) scores each candidate demonstration by marginal task information per token, (ii) penalizes redundancy and conflicts against the

current task hypothesis, (iii) greedily selects demonstrations under the budget, and (iv) compacts the selected context with a bounded predictive-divergence constraint.

## 3.1. Preliminary

A demonstration is $d_i = (x_i, y_i)$, with input $x_i$ and target output $y_i$. A context $C$ is a formatted concatenation of demonstrations, denoted by $\oplus$. A frozen (multi)modal model defines the conditional distribution

$$p_\theta(y \mid x, C), \qquad (1)$$

where $\theta$ is fixed, $x$ is an input, $y$ is an output, and $C$ is the in-context prompt.

Let $L(\cdot)$ denote token length under the model tokenizer, and let the demo cost be $\ell_i \triangleq L(d_i)$. We enforce a strict budget

$$L(C) \leq B. \qquad (2)$$

We fix the probe set as $\mathcal{X}_{\text{probe}} = \{x_q\}$.

## 3.2. Rate–Distortion View and Discrete Surrogate

ICL improves when additional tokens increase task information and degrades when they mostly add redundancy or conflicts. Rate–distortion formalizes this behavior by trading an information term against a distortion term under a token budget.

Let $T$ be a latent task variable with prior $p(T)$, and let $(X, Y) \sim p(\cdot \mid T)$ be task-conditioned examples. A context $C$ is a finite message about $T$ consumed by the predictor $p_\theta(y \mid x, C)$. We write the rate–distortion Lagrangian

$$\min_{p(C|T)} \; \mathbb{E}[\ell(Y, p_\theta(\cdot \mid X, C))] \; + \; \beta \, I(T; C) \; + \; \gamma \, \mathbb{E}[L(C)], \qquad (3)$$

where $\ell(\cdot)$ is predictive loss, $I(T; C)$ is mutual information under $p(T)p(C \mid T)$, and $\beta, \gamma \geq 0$ control the trade-off and token price. We restrict $C$ to be built from a subset $S \subseteq [N]$ of candidate demonstrations and approximate the incremental information and distortion terms using query-anchored predictive shift and prefix-conditioned conflict penalties.

## 3.3. Marginal Information Density

The information term in (3) increases when adding a demonstration changes the model's query prediction in a task-consistent way. Under a strict budget, a demonstration is valuable when this marginal effect is large per token.

Let $C(S) \triangleq \bigoplus_{j \in S} d_j$ be the formatted prefix context. We compute predictive shift on a finite answer set $\mathcal{Y}_q$ and define a categorical distribution

$$q_\theta(y \mid x_q, C) \propto \exp\big(s_\theta(y \mid x_q, C)\big), \qquad (4)$$

$$s_\theta(y \mid x_q, C) \triangleq \log p_\theta(y \mid x_q, C). \qquad (5)$$

for $y \in \mathcal{Y}_q$. Appendix A.1 specifies $\mathcal{Y}_q$ for each task family. The marginal task information of adding $d_i$ to prefix $S$ is the KL divergence

$$\text{IG}_i(S) \triangleq D_{\text{KL}}\Big(q_\theta(\cdot \mid x_q, C(S) \oplus d_i) \,\big\|\, q_\theta(\cdot \mid x_q, C(S))\Big). \qquad (6)$$

In (6), $D_{\text{KL}}(p\|q) = \sum_y p(y) \log \frac{p(y)}{q(y)}$ is computed on $\mathcal{Y}_q$, and $\text{IG}_i(S)$ measures the query-anchored predictive shift induced by $d_i$.

We downweight incoherent demonstrations using a reliability weight

$$w_i \triangleq \exp\Big(\tfrac{1}{L(y_i)} \log p_\theta(y_i \mid x_i, C_0)\Big) \in (0, 1], \qquad (7)$$

where $C_0$ is a fixed neutral header. The information density is

$$\rho_i(S) \triangleq \frac{w_i \, \text{IG}_i(S)}{\ell_i}. \qquad (8)$$

In (8), $\ell_i = L(d_i)$ is token cost, $w_i$ controls noise sensitivity, and $\rho_i(S)$ is marginal task information per token.

## 3.4. Redundancy Similarity and Conflict Cost

Redundant demonstrations waste tokens without increasing task information, while conflicting demonstrations increase distortion by injecting inconsistent task signals.

**Redundancy similarity.** We compute a fixed embedding $e_i \in \mathbb{R}^m$ for each demonstration $d_i$ from frozen-model hidden states. Redundancy is measured by cosine similarity:

$$s_{ij} \triangleq \frac{\langle e_i, e_j \rangle}{\|e_i\|_2 \, \|e_j\|_2} \in [-1, 1]. \qquad (9)$$

In (9), larger $s_{ij}$ indicates higher overlap and hence higher redundancy.

**Conflict cost.** Given prefix $S$, we measure how well $d_i = (x_i, y_i)$ fits the task hypothesis induced by $C(S)$ using per-token negative log-likelihood:

$$c_i(S) \triangleq -\frac{1}{L(y_i)} \log p_\theta\big(y_i \mid x_i, C(S)\big). \qquad (10)$$

In (10), larger $c_i(S)$ indicates that $d_i$ is less consistent with the current prefix.

## 3.5. Greedy Budgeted Selection

Selecting demonstrations under (2) is combinatorial. A greedy policy that maximizes marginal gain per token yields a simple solver aligned with the surrogate rate–distortion terms.

At prefix set $S$, the marginal gain of adding candidate $i \notin S$ is

$$\Delta(i \mid S) \triangleq \rho_i(S) \; - \; \lambda \sum_{j \in S} s_{ij} \; - \; \alpha \, c_i(S), \qquad (11)$$

where $\lambda \geq 0$ penalizes redundancy similarity, $\alpha \geq 0$ penalizes conflicts, $\rho_i(S)$ is defined in (8), $s_{ij}$ is defined in (9), and $c_i(S)$ is defined in (10). RDCO iteratively selects

$$i^\star \in \arg \max_{i \notin S, \ \ell_i \leq b} \Delta(i \mid S)/\ell_i \qquad (12)$$

with remaining budget $b$, updates $S \leftarrow S \cup \{i^\star\}$ and $b \leftarrow b - \ell_{i^\star}$, and stops when the best feasible marginal gain is non-positive. The final context is constructed from compacted demonstrations $\{\tilde{d}_i\}_{i \in S}$ produced by a bounded-divergence compaction operator:

$$C^\star \triangleq \bigoplus_{i \in S} \tilde{d}_i, \qquad \hat{y} \sim p_\theta(\cdot \mid x_q, C^\star). \qquad (13)$$

## 4. Theory

### 4.1. Token–Error Limits via Rate–Distortion

#### 4.1.1. A Stylized ICL Model for Generalization

We analyze an idealized ICL setting where the *task* is a latent parameter vector $\theta \in \mathbb{R}^d$ drawn from a Gaussian prior $\theta \sim \mathcal{N}(0, \tau^2 I_d)$. Conditioned on $\theta$, each demonstration is generated as

$$x \sim \mathcal{N}(0, I_d), \qquad y = x^\top \theta + \varepsilon, \qquad \varepsilon \sim \mathcal{N}(0, \sigma^2), \qquad (14)$$

independently across samples. A candidate pool of $N$ demonstrations is $\mathcal{D}_N \triangleq \{(x_i, y_i)\}_{i=1}^N$. A query $(x_q, y_q)$ is drawn from the same mechanism (14). A context is a token sequence $C$ produced from $\mathcal{D}_N$ with token budget $B$:

$$C = \mathrm{Enc}(\mathcal{D}_N) \in \mathcal{V}^B, \qquad (15)$$

where $\mathcal{V}$ is a vocabulary of size $|\mathcal{V}| = V$ (tokens), and $\mathrm{Enc}$ is any (possibly randomized) encoder. Given $C$ and $x_q$, a predictor outputs $\hat{y}(x_q, C)$. We measure the (Bayes) generalization error by the squared loss

$$\mathcal{R}(C) \triangleq \mathbb{E}\big[(y_q - \hat{y}(x_q, C))^2\big], \qquad (16)$$

where the expectation is over $\theta$, $\mathcal{D}_N$, and the query $(x_q, y_q)$. All logarithms are natural.

> Main Result: RD Lower Bound and RDCO Upper Bound

**Theorem 4.1** (Token–Error Lower Bound and RDCO Upper Bound). *Consider the model* (14)–(16).

*(**Lower bound**). For any encoder $C = \mathrm{Enc}(\mathcal{D}_N) \in \mathcal{V}^B$ and any predictor $\hat{y}(x_q, C)$,*

$$\mathcal{R}(C) \geq \sigma^2 + d\tau^2 \exp\Big(-\tfrac{2}{d} R_{\mathrm{eff}}(B, N)\Big), \qquad (17)$$

*where the effective rate satisfies*

$$R_{\mathrm{eff}}(B, N) \triangleq \min\big\{B \log V, \ I(\theta; \mathcal{D}_N)\big\}. \qquad (18)$$

*(**Upper bound for RDCO under regularity**). Fix an integer $k$ and assume the encoder constructs $C$ by selecting $k$ demonstrations and serializing them in-context, with $k$ chosen such that the token budget $B$ allows exactly $k$ formatted demonstrations.[1] Let $S \subseteq [N]$ denote the selected indices, and define the information set function*

$$F(S) \triangleq I(\theta; y_S \mid X_S) = \tfrac{1}{2} \log \det\Big(I_d + \tfrac{\tau^2}{\sigma^2} X_S^\top X_S\Big), \qquad (19)$$

*where $X_S \in \mathbb{R}^{k \times d}$ stacks $\{x_i^\top\}_{i \in S}$ and $y_S$ stacks $\{y_i\}_{i \in S}$. Let $S^\star \in \arg \max_{|S|=k} F(S)$ be an optimal size-$k$ subset, and let $S_g$ be the greedy subset that iteratively maximizes the marginal gain of $F$. Assume the selected design is well-conditioned in the sense that*

$$(1-\delta) k I_d \preceq X_{S_g}^\top X_{S_g} \preceq (1+\delta) k I_d \quad \text{for some } \delta \in (0, 1). \qquad (20)$$

*Then the Bayes-optimal predictor given $C(S_g)$ satisfies*

$$\mathcal{R}\big(C(S_g)\big) \leq \sigma^2 + \kappa(\delta, k) d\tau^2 \exp\Big(-\tfrac{2}{d}\big(1 - \tfrac{1}{e}\big) F(S^\star)\Big), \qquad (21)$$

*where*

$$\kappa(\delta, k) \triangleq \frac{1 + \tfrac{\tau^2}{\sigma^2}(1 + \delta)k}{1 + \tfrac{\tau^2}{\sigma^2}(1 - \delta)k}. \qquad (22)$$

*Moreover, the greedy selection achieves the approximation guarantee*

$$F(S_g) \geq \Big(1 - \tfrac{1}{e}\Big) F(S^\star). \qquad (23)$$

**Proof sketch (Theorem 4.1).** For the lower bound, we (i) relate prediction error to parameter error via $\mathbb{E}[(x_q^\top(\theta - \hat{\theta}(C)))^2] = \mathbb{E}\|\theta - \hat{\theta}(C)\|_2^2$ under $x_q \sim \mathcal{N}(0, I_d)$, (ii) apply the Gaussian rate–distortion converse to obtain $\mathbb{E}\|\theta - \hat{\theta}(C)\|_2^2 \geq d\tau^2 \exp\big(-\tfrac{2}{d} I(\theta; C)\big)$, and (iii) use data processing $I(\theta; C) \leq I(\theta; \mathcal{D}_N)$ and the entropy bound $I(\theta; C) \leq H(C) \leq B \log V$ to get (17)–(18). For the upper bound, we (i) compute $F(S)$ in (19) for the linear-Gaussian model, (ii) show $F$ is monotone submodular, yielding the classical greedy bound (23), (iii) express the Bayes risk as $\sigma^2 + \tau^2 \mathrm{tr}\big(I_d + \tfrac{\tau^2}{\sigma^2} X_{S_g}^\top X_{S_g}\big)^{-1}$, and (iv) bound the trace via the spectral sandwich (20) and relate it to $\exp(-2F(S_g)/d)$, giving (21) with factor (22). Full proofs are in Appendix B.1.

#### 4.1.2. Token–Sample Complexity Implication

Theorem 4.1 yields an explicit token–sample trade-off in this stylized model.

**Corollary 4.2** (Token Complexity for Excess Error $\varepsilon$). *Under the assumptions of Theorem 4.1 and the conditioning*

---

[1] This corresponds to a fixed per-demo prompt template length.

(20), *if $k$ demonstrations fit in the budget and the greedy-selected context is used, then*

$$\mathcal{R}\big(C(S_g)\big) - \sigma^2 \ \leq \ \frac{d\tau^2}{1 + \frac{\tau^2}{\sigma^2}(1-\delta)k}. \qquad (24)$$

*In particular, to achieve $\mathcal{R}(C(S_g)) - \sigma^2 \leq \varepsilon$, it suffices that*

$$k \ \geq \ \frac{\sigma^2}{\tau^2} \cdot \frac{d}{(1-\delta)\varepsilon}. \qquad (25)$$

## 4.2. Approximation Guarantees for Budgeted Context Selection

Objective: Information Gain with Redundancy and Conflict Penalties.

Let the ground set be $\mathcal{N} \triangleq \{1, 2, \ldots, N\}$ indexing candidate demonstrations. We analyze subset selection under a *cardinality* budget $k$ (corresponding to a fixed per-demo token template after compaction). We consider the following utility, which matches the greedy marginal used by our selector:

$$G(S) \ \triangleq \ F(S) - \lambda P(S) - \alpha M(S), \qquad S \subseteq \mathcal{N}, \ (26)$$

where: (i) $F(S)$ is a nonnegative *task-information* set function (e.g., mutual information / log-determinant gain); (ii) $P(S)$ is a *redundancy* penalty based on pairwise similarities; (iii) $M(S)$ is a *conflict* penalty (modular in this section). We define

$$P(S) \ \triangleq \ \sum_{\substack{i,j \in S \\ i < j}} s_{ij}, \qquad M(S) \ \triangleq \ \sum_{i \in S} m_i, \qquad (27)$$

where $s_{ij} = s_{ji} \geq 0$ is a fixed similarity score and $m_i \geq 0$ is a fixed per-demo conflict score.

We study the greedy algorithm that starts from $S_0 = \emptyset$ and iteratively adds

$$\begin{aligned} i_t &\in \arg\max_{i \in \mathcal{N} \setminus S_t} \Delta_G(i \mid S_t), \\ \Delta_G(i \mid S) &\triangleq G(S \cup \{i\}) - G(S), \end{aligned} \qquad (28)$$

until $|S_t| = k$, producing $S_g \triangleq S_k$.

## 4.3. Submodularity and Greedy Approximation

**Theorem 4.3** (Submodular Structure and Greedy Approximation). *Assume $F$ is normalized ($F(\emptyset) = 0$), nonnegative, and monotone submodular. Let $P$ and $M$ be defined in (27) with $s_{ij} \geq 0$ and $m_i \geq 0$, and define $G$ by (26). Then:*

*(i) Submodularity. The utility $G$ is submodular.*

*(ii) Monotonicity under a margin condition. Define the $k$-step marginal floor of $F$ as*

$$\mu_k \ \triangleq \ \min_{\substack{S \subseteq \mathcal{N}, \ |S| \leq k-1 \\ i \in \mathcal{N} \setminus S}} \big(F(S \cup \{i\}) - F(S)\big). \qquad (29)$$

*Let $s_{\max} \triangleq \max_{i \neq j} s_{ij}$ and $m_{\max} \triangleq \max_i m_i$. If*

$$\mu_k \ \geq \ \lambda\,(k-1)\,s_{\max} \ + \ \alpha\,m_{\max}, \qquad (30)$$

*then $G$ is monotone on all sets of size at most $k$.*

*(iii) Greedy guarantee.* *If (30) holds, then for the cardinality-constrained maximizer*

$$S^\star \ \in \ \arg\max_{S \subseteq \mathcal{N}: \ |S| \leq k} G(S), \qquad (31)$$

*the greedy set $S_g$ from (28) satisfies*

$$G(S_g) \ \geq \ \Big(1 - \frac{1}{e}\Big) G(S^\star). \qquad (32)$$

*(iv) Weak-submodular extension. If instead of submodularity, $G$ satisfies the order-$k$ weak submodularity inequality*

$$\sum_{i \in L} \Delta_G(i \mid S) \ \geq \ \gamma_k \, \Delta_G(L \mid S)$$

$$\text{for all } S \subseteq \mathcal{N}, \ L \subseteq \mathcal{N} \setminus S, \ |L| \leq k, \quad (33)$$

*for some $\gamma_k \in (0, 1]$, and $G$ is monotone, then greedy satisfies*

$$G(S_g) \ \geq \ \big(1 - e^{-\gamma_k}\big) G(S^\star). \qquad (34)$$

**Proof sketch (Theorem 4.3).** (i) We show $P$ is supermodular because its marginal gain when adding $i$ equals $\sum_{j \in S} s_{ij}$, which increases with $S$; hence $-\lambda P$ is submodular. Since $F$ is submodular and $M$ is modular, their sum (26) is submodular. (ii) For any $|S| \leq k - 1$ and $i \notin S$, we lower bound the marginal gain $\Delta_G(i \mid S) = \Delta_F(i \mid S) - \lambda \sum_{j \in S} s_{ij} - \alpha m_i$ by $\mu_k - \lambda(k-1)s_{\max} - \alpha m_{\max}$, which is nonnegative under (30), implying monotonicity. (iii) Under monotone submodularity, the standard greedy analysis yields an exponential residual decay: $G(S^\star) - G(S_{t+1}) \leq (1 - \frac{1}{k})(G(S^\star) - G(S_t))$; iterating gives (32). (iv) Replacing submodularity by (33) yields the same contraction with factor $\gamma_k/k$, producing (34). Full proofs appear in Appendix B.2.

# 5. Experiments

## 5.1. Experimental Setup

**Datasets.** We evaluate our method on widely used ICL suites drawn from recent SOTA/baseline papers, covering (i) **demonstration selection** across mixed task types, (ii) **calibration / stability** under dynamic contexts, and (iii) **OOD robustness** and **retrieval-augmented QA**. Specifically, we include: *(A)* the 10-dataset multi-task benchmark of Liu et al. (2025), *(B)* the 8-dataset dynamic-calibration benchmark of Tan et al. (2025), *(C)* the OOD retrieval benchmarks used by Wang et al. (2024) and Byun et al. (2026), *(D)* open-domain QA with contexts from Kahardipraja et al.

*Table 1.* **Main results.** Best/2nd/3rd are shaded.

*(a)* **Classification (Acc., %).**

| Method | SST-5 | MRPC | QNLI | CMSQA | SWAG | Avg. |
|---|---|---|---|---|---|---|
| Random | 28.61 | 65.93 | 55.08 | 42.34 | 41.39 | 46.67 |
| Top-K BM25 | 32.06 | 65.93 | 60.11 | 35.79 | 43.35 | 47.45 |
| Top-K BERT | 32.70 | 69.12 | 60.94 | 35.87 | 41.09 | 47.94 |
| Top-K SBERT | 39.50 | 70.34 | 60.46 | 31.53 | 40.92 | 48.55 |
| MLSM | 33.15 | 69.87 | 65.02 | 37.26 | 41.49 | 49.36 |
| EPR | 36.88 | 81.37 | 77.87 | 38.74 | 43.39 | 55.65 |
| CEIL | 37.69 | 77.94 | 80.58 | 38.90 | 43.84 | 55.79 |
| LLM-R | 36.55 | 75.23 | 78.30 | 36.81 | 43.17 | 54.18 |
| RLDES | 37.24 | 76.86 | 79.52 | 37.51 | 44.26 | 55.04 |
| CCL | 36.85 | 77.12 | 78.95 | 38.20 | 43.61 | 54.92 |
| CC | 34.54 | 73.20 | 71.32 | 36.12 | 42.85 | 51.58 |
| BC | 35.27 | 74.63 | 72.88 | 36.88 | 43.20 | 52.52 |
| LinC | 35.80 | 75.10 | 73.50 | 37.31 | 43.52 | 53.04 |
| SC | 36.23 | 75.80 | 74.22 | 37.60 | 43.72 | 53.50 |
| TTF | 42.14 | 74.51 | 85.08 | 47.83 | 55.72 | 61.06 |
| RDCO (Ours) | 41.75 | 82.60 | 86.20 | 48.95 | 56.80 | 63.26 |

*(b)* **Generation (EM, %).**

| Method | WebQs | GeoQ. | NL2B. | MTOP | SMCal. | Avg. |
|---|---|---|---|---|---|---|
| Random | 3.79 | 25.36 | 31.27 | 3.98 | 3.70 | 13.62 |
| Top-K BM25 | 14.17 | 65.71 | 58.81 | 49.66 | 44.02 | 46.47 |
| Top-K BERT | 14.17 | 64.64 | 52.45 | 51.36 | 44.76 | 45.48 |
| Top-K SBERT | 15.15 | 60.71 | 46.87 | 46.80 | 42.79 | 42.46 |
| MLSM | 16.14 | 68.93 | 56.11 | 54.05 | 47.72 | 48.59 |
| EPR | 17.62 | 73.21 | 77.87 | 60.82 | 60.49 | 58.00 |
| CEIL | 17.08 | 70.71 | 53.66 | 63.40 | 56.30 | 52.23 |
| LLM-R | 16.80 | 71.56 | 75.25 | 59.81 | 58.63 | 56.41 |
| RLDES | 16.95 | 72.33 | 76.50 | 60.56 | 59.20 | 57.11 |
| CCL | 17.06 | 72.82 | 76.88 | 60.88 | 59.52 | 57.43 |
| CC | 15.52 | 68.24 | 70.37 | 57.40 | 55.26 | 53.36 |
| BC | 15.85 | 69.11 | 71.26 | 58.12 | 56.37 | 54.14 |
| LinC | 16.13 | 69.87 | 72.10 | 58.91 | 57.02 | 54.81 |
| SC | 16.42 | 70.56 | 73.23 | 59.51 | 57.87 | 55.52 |
| TTF | 17.07 | 71.43 | 46.30 | 58.12 | 51.06 | 48.80 |
| RDCO (Ours) | 18.95 | 75.68 | 79.12 | 65.84 | 61.73 | 60.26 |

(2026), *(E)* long-context and prompt-position sensitivity tasks from Cobbina & Zhou (2025), and *(F)* intent classification and challenging reasoning benchmarks used by Wang et al. (2025). Appendix C.1 enumerates all datasets, task types, and evaluation splits.

**Evaluation metrics.** We follow each suite's standard metrics. For *classification*, we report *Accuracy* (Acc.). For *structured generation* and *semantic parsing*, we report *Exact Match* (EM) following Liu et al. (2025). For *open-domain QA*, we report *Recall* (primary) and also EM / semantic match when applicable, following Kahardipraja et al. (2026). For *summarization*, we report *ROUGE-L* following Cobbina & Zhou (2025). To quantify *token efficiency*—the core objective of our method—we additionally report: (i) $Acc@B$ (performance at a fixed token budget $B$) and (ii) $Tok@Acc$ (minimum tokens required to match a target accuracy, computed by sweeping budgets). For *stability*, we adopt *Accuracy-Change* and *Prediction-Change* as in Cobbina & Zhou (2025). For *calibration*, we report *NLL* and *ECE* following Zhao et al. (2021); Zhou et al. (2024); Abbas et al. (2024); Tan et al. (2025). Appendix C.2 provides exact definitions and extraction rules per task.

**Compared methods.** We compare against recent ICL demonstration selection and calibration baselines, grouped by *training requirement* for fair comparison. **Training-free selection/retrieval** baselines include Random, BM25 retrieval (Robertson et al., 2009), dense Top-$K$ retrieval with BERT/SBERT embeddings (Devlin et al., 2019; Reimers & Gurevych, 2019), and DPP-style compositional subset selection (Ye et al., 2023). **Learning-based selection** baselines include EPR (Rubin et al., 2022), LLM-R (Wang et al., 2024), and RL-based selection (Wang et al., 2025), as well as causal representation based OOD selection (Byun et al.,

2026). **Calibration** baselines include Contextual Calibration (CC) (Zhao et al., 2021), Batch Calibration (BC) (Zhou et al., 2024), Linear Probe Calibration (LinC) (Abbas et al., 2024), and Surprise Calibration (SC) (Tan et al., 2025). When a baseline optimizes a different axis (e.g., calibration rather than selection), we apply it *on top of* a shared retrieval pool and the same backbone model, so that differences reflect the targeted component rather than confounded prompt formats. Appendix C.3 details the exact settings and which baselines are evaluated on which suites.

**Implementation details.** We use instruction-tuned backbones commonly adopted in recent ICL studies (Kahardipraja et al., 2026; Cobbina & Zhou, 2025; Tan et al., 2025), and keep the backbone *frozen* for all methods. Prompts place demonstrations at the start of the system prompt (ssp) to avoid demo-position bias (Cobbina & Zhou, 2025; Zang & Liu, 2024). For each query, we first form a candidate pool via BM25 or dense retrieval, then run greedy budgeted selection under token budget $B$ and compaction to satisfy $L(C) \leq B$. Hyperparameter grids are reported in Appendix C.4.

### 5.2. Main Results

We compare RDCO against widely adopted demonstration-selection baselines under the standard 20-shot ICL protocol on ten datasets spanning classification and generation tasks.

**Classification tasks.** Table 1a reports Accuracy on five classification-style benchmarks. Overall, supervised retriever training (EPR/CEIL/TTF) substantially outperforms unsupervised similarity retrieval. Among baselines, TTF achieves the strongest average accuracy (61.06), while RDCO achieves the best overall classification average

(63.26), improving over TTF by +2.20 points. RDCO is evaluated under the same demonstration budget and prompt template.

**Generation tasks.** Table 1b reports Exact Match (EM) on five generation/semantic parsing benchmarks. Among reported baselines, EPR provides the strongest average EM (58.00), while CCL (57.43) and RLDES (57.11) are the next strongest baselines on average; CEIL remains the strongest baseline on MTOP. This gap highlights that learning-based retrievers are powerful but costly to train/maintain.

### 5.3. Ablations and Analysis

**Single-factor ablations.** Single-factor ablations in Table 2 show removing *marginal* information gain causes the largest drop (**Overall** $-2.49$), with CMSQA ($-2.60$) and MTOP ($-3.53$) most affected—confirming context-dependent marginal utilities are essential to avoid redundant evidence. Length normalization is next most critical (**Overall** $-1.30$), showing *information density* allocation prevents long demonstrations from dominating coverage. Redundancy and conflict control both matter: removing redundancy hurts SWAG and CMSQA ($-1.38, -1.37$), while dropping conflict penalty especially harms generation (Gen. Avg. $-1.67$), as mismatched demonstrations inject inconsistent signals. KL-bounded compaction improves results under fixed budget (Gen. Avg. $-1.59$ drop without it), notably on MTOP and SMCal. ($\sim -1.9$ each), indicating controlled compression preserves task content while freeing tokens for additional evidence.

**Redundancy and Conflict Penalties.** We diagnose whether RDCO is controllable via the redundancy penalty $\lambda$ and conflict penalty $\alpha$ (Section 3.5), and whether there exists a stable optimum basin rather than a brittle, single-point optimum. This directly tests the rate–distortion claim that performance degrades when tokens are dominated by redundancy (high $\lambda$ needed) or injected conflicts (high $\alpha$ needed), but over-penalization can also suppress useful evidence. We fix the token budget to $B=256$ and keep the candidate pool construction identical across methods (same retriever, same pool size). We run a 2D grid sweep over $(\lambda, \alpha)$: $\lambda \in \{0.00, 0.04, \ldots, 1.20\}$ and $\alpha \in \{0.00, 0.05, \ldots, 2.00\}$. For each pair $(\lambda, \alpha)$, we evaluate RDCO on the validation split.

Figure 2 exhibits a broad high-utility basin (rather than a needle-in-a-haystack optimum), suggesting RDCO is not brittle to modest hyperparameter perturbations. When $\lambda \approx 0$, performance drops due to near-duplicate selection (wasting budget on redundant evidence), while overly large $\lambda$ suppresses coverage and hurts overall accuracy. Similarly, $\alpha$ improves robustness by filtering inconsistent demonstrations, but overly large $\alpha$ becomes conservative and discards informative but slightly mismatched evidence, degrading

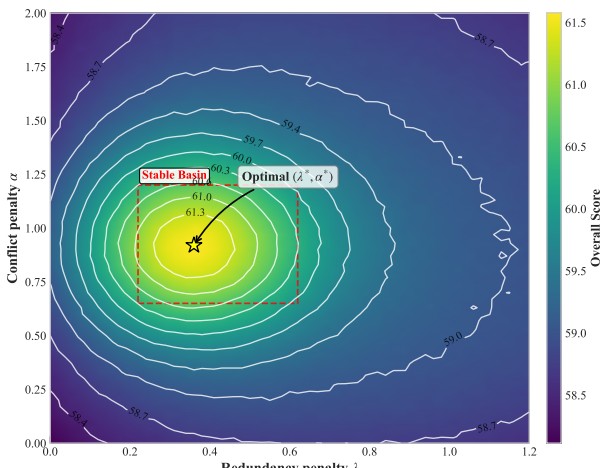

*Figure 2.* **Phase diagram of RDCO over** $(\lambda, \alpha)$**.** Heatmap shows the Overall score under a fixed token budget ($B=256$) and fixed candidate pool; higher is better. Contours mark equal-score level sets, revealing a broad optimum basin. The star indicates the selected hyperparameters $(\lambda^\star, \alpha^\star)$ used in all other experiments.

performance. We therefore choose $(\lambda^\star, \alpha^\star)$ near the basin center to maximize both performance and stability.

**Greedy Selection Trajectory.** A core claim of RDCO is a rate–distortion behavior: early demonstrations add high marginal task information per token, but as the context grows, additional tokens increasingly encode redundancy and conflicts, yielding a diminishing (and eventually non-positive) marginal gain. We test this claim by visualizing the trajectory of greedy selection in the space of cumulative redundancy and cumulative information, with cumulative conflict encoded in marker size. Detailed experimental settings are provided in §C.5.

Fig. 3 reveals a consistent two-regime behavior that matches the rate–distortion narrative. Early steps climb steeply in $I_t$ with modest growth in $R_t$ (tokens are information-efficient), while later steps drift rightward with shrinking vertical progress (additional demonstrations are increasingly redundant). Simultaneously, marker sizes tend to grow in late steps, indicating that remaining candidates are more conflict-prone, which explains why the best marginal gain $g_t$ eventually becomes non-positive and the greedy procedure halts. Overall, the observed "bend then flatten" trajectories provide direct process-level evidence that RDCO spends budget on informative tokens first and naturally avoids the redundancy/conflict-dominated tail of the candidate pool.

**KL-Bounded Compaction.** Our compaction operator produces compacted demonstrations $\{\tilde{d}_i\}$ such that the final context fits a strict token budget (Eq. 2) while controlling predictive distortion via a bounded divergence constraint. We may ask whether compaction is merely truncation in disguise, and whether the KL bound actually prevents output

*Table 2.* **Single-factor ablations on the 10-dataset suite (per-dataset).** Classification uses Acc. (%), generation uses EM (%).

| | Classification (Acc., %) | | | | | | Generation (EM, %) | | | | | | Overall |
|---|---|---|---|---|---|---|---|---|---|---|---|---|---|
| Variant | SST-5 | MRPC | QNLI | CMSQA | SWAG | Avg. | WebQs | GeoQ. | NL2B. | MTOP | SMCal. | Avg. | |
| RDCO (Full) | 41.75 | 82.60 | 86.20 | 48.95 | 56.80 | 63.26 | 18.95 | 75.68 | 79.12 | 65.84 | 61.73 | 60.26 | 61.76 |
| w/o marginal IG | 40.02(-1.73) | 80.85(-1.75) | 84.01(-2.19) | 46.35(-2.60) | 54.91(-1.89) | 61.23(-2.03) | 16.79(-2.16) | 72.66(-3.02) | 76.48(-2.64) | 62.31(-3.53) | 58.31(-3.42) | 57.31(-2.95) | 59.27(-2.49) |
| w/o length normalization | 40.58(-1.17) | 81.63(-0.97) | 85.02(-1.18) | 47.66(-1.29) | 55.89(-0.91) | 62.16(-1.10) | 17.96(-0.99) | 74.21(-1.47) | 77.32(-1.80) | 64.20(-1.64) | 60.11(-1.62) | 58.76(-1.50) | 60.46(-1.30) |
| w/o reliability weight $w_i$ | 41.12(-0.63) | 81.90(-0.70) | 85.54(-0.66) | 48.15(-0.80) | 56.17(-0.63) | 62.58(-0.68) | 18.21(-0.74) | 74.93(-0.75) | 78.05(-1.07) | 65.08(-0.76) | 60.62(-1.11) | 59.38(-0.88) | 60.98(-0.78) |
| w/o redundancy penalty ($\lambda{=}0$) | 40.44(-1.31) | 81.72(-0.88) | 85.21(-0.99) | 47.58(-1.37) | 55.42(-1.38) | 62.07(-1.19) | 18.03(-0.92) | 74.56(-1.12) | 77.88(-1.24) | 64.76(-1.08) | 60.70(-1.03) | 59.19(-1.07) | 60.63(-1.13) |
| w/o conflict penalty ($\alpha{=}0$) | 41.01(-0.74) | 81.55(-1.05) | 85.69(-0.51) | 48.03(-0.92) | 56.06(-0.74) | 62.47(-0.79) | 17.62(-1.33) | 73.90(-1.78) | 77.41(-1.71) | 64.05(-1.79) | 59.98(-1.75) | 58.59(-1.67) | 60.53(-1.23) |
| w/o KL-bounded compaction ($\varepsilon{=}0$) | 40.88(-0.87) | 81.86(-0.74) | 85.44(-0.76) | 47.93(-1.02) | 55.94(-0.86) | 62.41(-0.85) | 17.54(-1.41) | 74.10(-1.58) | 77.95(-1.17) | 63.92(-1.92) | 59.82(-1.91) | 58.67(-1.59) | 60.54(-1.22) |

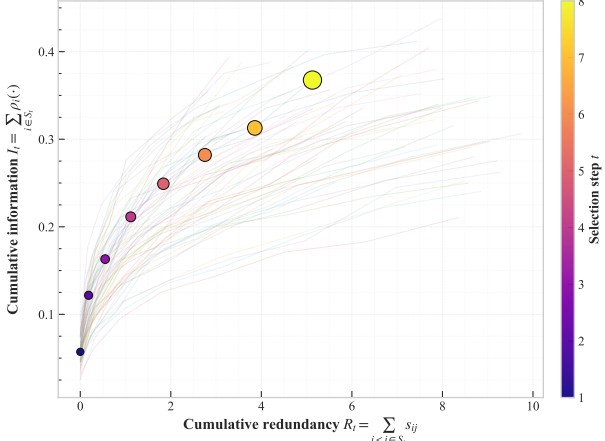

*Figure 3.* **Greedy selection trajectory of RDCO.** Each polyline is one query's greedy selection path in the $(R_t, I_t)$ plane, where $R_t$ is cumulative redundancy and $I_t$ is cumulative information density. A representative trajectory is overlaid with step-colored markers; marker size encodes cumulative conflict $K_t$. The trajectory typically exhibits an early *high-information/low-redundancy* phase followed by a *saturation* phase where redundancy and conflicts accumulate faster than information, producing a visible bend/flattening before stopping ($g_t \leq 0$).

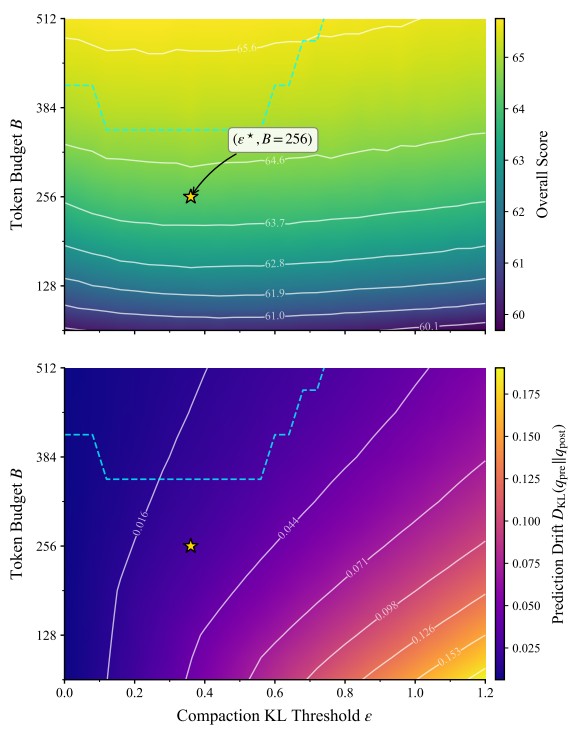

*Figure 4.* **Compression–drift phase diagram over** $(\varepsilon, B)$. **Left:** Overall score as a function of divergence threshold $\varepsilon$ and token budget $B$ (higher is better). **Right:** Prediction drift $\mathrm{Drift}(\varepsilon, B)$ from Eq. 143 (lower is better). The dashed contour highlights a region that is simultaneously near-optimal in score and low in drift, showing that the KL-bounded compaction admits a *high-performance / low-drift* operating regime.

drift. We therefore map the joint effect of the divergence threshold $\varepsilon$ and the token budget $B$ on both task performance and prediction drift. Details are provided in §C.6.

Figure 4 shows a clear *operating sweet spot*. When $\varepsilon$ is too small, compaction is overly conservative: under tight budgets (small $B$), the context cannot be compacted enough to pack complementary evidence, and the score saturates early. As $\varepsilon$ increases to a moderate range, score improves markedly (especially at smaller $B$), indicating that bounded compaction frees tokens for additional, non-redundant demonstrations. Crucially, the drift heatmap confirms that this gain is achieved without large predictive shifts: a broad region of high score overlaps with low drift, consistent with the intended bounded-divergence behavior. Finally, when $\varepsilon$ becomes too permissive, drift rises rapidly and the score declines, revealing a concrete *trade-off boundary* beyond which compression begins to distort task-relevant signals. Overall, this phase diagram provides direct evidence that RDCO's compaction is not "truncation for free" but a con-

trollable mechanism that can be tuned to jointly optimize token efficiency and prediction stability.

## 6. Conclusion

We studied token-budgeted context construction for in-context learning, where prompts must add task signal without amplifying redundancy or conflicts. We proposed RDCO, a training-free optimizer that selects demonstrations by marginal information per token and compacts them while controlling predictive drift. Across diverse ICL benchmarks, RDCO improves token efficiency and delivers a better performance–cost frontier than competitive baselines.

## Impact Statement

This work aims to improve the efficiency and robustness of in-context learning under limited context budgets. Potential risks include over-reliance on automatically selected demonstrations and amplification of biases present in the candidate pool. In deployment, these risks should be mitigated through retrieval-pool auditing, calibration monitoring, and task-specific safety filters.

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

## Appendix Contents

## A. Additional Method Details

### A.1. Answer Set $\mathcal{Y}_q$ and Normalized Distribution

The KL proxy in (6) is computed on a finite answer set $\mathcal{Y}_q$.

**Closed-set prediction.** For closed-set tasks, $\mathcal{Y}_q$ equals the label set.

**Open-set generation.** For open-set generation, we define $\mathcal{Y}_q$ as a deterministic candidate set generated under the neutral header $C_0$. Let $\mathrm{Gen}_K(p_\theta, x_q, C_0)$ return the top-$K$ completed sequences under a fixed decoding rule, and set

$$\mathcal{Y}_q \triangleq \mathrm{Gen}_K(p_\theta, x_q, C_0). \tag{35}$$

We then define

$$q_\theta(y \mid x_q, C) \triangleq \frac{\exp\big(s_\theta(y \mid x_q, C)\big)}{\sum_{y' \in \mathcal{Y}_q} \exp\big(s_\theta(y' \mid x_q, C)\big)}, \qquad s_\theta(y \mid x_q, C) \triangleq \log p_\theta(y \mid x_q, C), \tag{36}$$

which makes $D_{\mathrm{KL}}$ in (6) computable.

## A.2. Demonstration Embeddings

Let $u_{i,1:\ell_i}$ be the serialized token sequence of demonstration $d_i$ and let $h_\theta(u_{i,t}) \in \mathbb{R}^m$ be the final-layer hidden state of token $u_{i,t}$ under the frozen model. We use mean pooling:

$$e_i \triangleq \frac{1}{\ell_i} \sum_{t=1}^{\ell_i} h_\theta(u_{i,t}). \tag{37}$$

Cosine similarity $s_{ij}$ is then computed by (9). For multimodal models, the serialization includes fused visual tokens; (37) remains unchanged with $\ell_i = L(d_i)$ counting effective tokens.

## A.3. Bounded-Divergence Compaction

Given a selected demo $d_i$ and a fixed prefix $C(S \setminus \{i\})$, compaction constructs $\tilde{d}_i$ by minimizing token length while bounding the query-anchored predictive divergence:

$$\tilde{d}_i \triangleq \arg\min_{\tilde{d}} \ L(\tilde{d}) \quad \text{s.t.} \quad D_{\mathrm{KL}}\Big(q_\theta(\cdot \mid x_q, C(S \setminus \{i\}) \oplus \tilde{d}) \,\big\|\, q_\theta(\cdot \mid x_q, C(S \setminus \{i\}) \oplus d_i)\Big) \leq \varepsilon, \tag{38}$$

where $\varepsilon \geq 0$ is fixed.

**Atomization and pruning.** We serialize $d_i$ into ordered atoms $a_{i,1:M}$ and write $d_i = \bigoplus_{k=1}^{M} a_{i,k}$. For each atom, define its importance as the KL increase when removing it:

$$\mathrm{imp}(a_{i,k}) \triangleq D_{\mathrm{KL}}\Big(q_\theta(\cdot \mid x_q, C(S \setminus \{i\}) \oplus d_i) \,\big\|\, q_\theta(\cdot \mid x_q, C(S \setminus \{i\}) \oplus (d_i \setminus a_{i,k}))\Big). \tag{39}$$

Compaction iteratively removes the atom with smallest $\mathrm{imp}(a_{i,k})$ while maintaining (38).

## A.4. Complexity Notes

Embeddings $\{e_i\}$ and similarities $\{s_{ij}\}$ are computed once per pool. Reliability weights $w_i$ are computed once under $C_0$. Each greedy step computes $\mathrm{IG}_i(S)$ and $c_i(S)$ via forward evaluations of $p_\theta(\cdot \mid x, C(S))$ and $p_\theta(y_i \mid x_i, C(S))$, respectively. With pool size $N$ and selected size $K$, the greedy loop performs $O(KN)$ marginal evaluations.

# B. Proofs

## B.1. Full Proofs for Section 4.1

### B.1.1. PRELIMINARIES

**Lemma B.1** (Entropy and data-processing rate constraints)**.** *Let $C \in \mathcal{V}^B$ be any (possibly randomized) function of $\mathcal{D}_N$. Then*

$$I(\theta; C) \ \leq \ I(\theta; \mathcal{D}_N), \tag{40}$$

*and*

$$I(\theta; C) \ \leq \ H(C) \ \leq \ B \log V. \tag{41}$$

*Proof.* Since $C = \mathrm{Enc}(\mathcal{D}_N)$, we have the Markov chain $\theta \to \mathcal{D}_N \to C$. By data processing,

$$I(\theta; C) \ \leq \ I(\theta; \mathcal{D}_N), \tag{42}$$

which proves (40). For (41), mutual information is bounded by entropy:

$$I(\theta; C) \ \leq \ H(C). \tag{43}$$

Moreover, $C$ is a length-$B$ sequence over an alphabet of size $V$, hence

$$H(C) \ \leq \ \log |\mathcal{V}^B| \ = \ B \log V. \tag{44}$$

Combining (43) and (44) yields (41). $\qquad\square$

**Lemma B.2** (Prediction error reduces to parameter error). *Let $x_q \sim \mathcal{N}(0, I_d)$ be independent of $(\theta, C)$ and let $y_q = x_q^\top \theta + \varepsilon_q$ with $\varepsilon_q \sim \mathcal{N}(0, \sigma^2)$ independent. For any estimator $\widehat{y}(x_q, C)$, define $\widehat{\theta}(C)$ as the (possibly randomized) coefficient vector such that $\widehat{y}(x_q, C) = x_q^\top \widehat{\theta}(C)$.[2] Then*

$$\mathbb{E}\big[(y_q - \widehat{y}(x_q, C))^2\big] \;=\; \sigma^2 \;+\; \mathbb{E}\big[\|\theta - \widehat{\theta}(C)\|_2^2\big]. \tag{45}$$

*Proof.* Write

$$y_q - \widehat{y}(x_q, C) \;=\; x_q^\top (\theta - \widehat{\theta}(C)) + \varepsilon_q. \tag{46}$$

Since $\varepsilon_q$ is independent with mean zero and variance $\sigma^2$,

$$\mathbb{E}\big[(y_q - \widehat{y})^2\big] \;=\; \mathbb{E}\big[(x_q^\top (\theta - \widehat{\theta}(C)))^2\big] + \sigma^2. \tag{47}$$

Condition on $(\theta, C)$. Using $x_q \sim \mathcal{N}(0, I_d)$ independent,

$$\mathbb{E}\big[(x_q^\top (\theta - \widehat{\theta}(C)))^2 \mid \theta, C\big] \;=\; (\theta - \widehat{\theta}(C))^\top \mathbb{E}[x_q x_q^\top](\theta - \widehat{\theta}(C)) \;=\; \|\theta - \widehat{\theta}(C)\|_2^2. \tag{48}$$

Taking expectations in (48) and substituting into (47) yields (45). $\qquad\square$

### B.1.2. GAUSSIAN RATE–DISTORTION CONVERSE

**Lemma B.3** (Gaussian RD lower bound (vector, total MSE)). *Let $\theta \sim \mathcal{N}(0, \tau^2 I_d)$ and let $C$ be any random variable (arbitrary alphabet). For any estimator $\widehat{\theta}(C)$,*

$$\mathbb{E}\big[\|\theta - \widehat{\theta}(C)\|_2^2\big] \;\geq\; d\tau^2 \exp\Big(-\tfrac{2}{d} I(\theta; C)\Big). \tag{49}$$

*Equivalently, the rate–distortion function for total squared error is*

$$R(D) \;=\; \inf_{p(\widehat{\theta}|\theta):\, \mathbb{E}\|\theta - \widehat{\theta}\|_2^2 \leq D} I(\theta; \widehat{\theta}) \;=\; \frac{d}{2} \log\Big(\frac{d\tau^2}{D}\Big), \quad 0 < D \leq d\tau^2. \tag{50}$$

*Proof.* Let the estimation error be $E \triangleq \theta - \widehat{\theta}(C)$. We use a standard entropy-power / maximum-entropy argument.

**Step 1: upper bound $h(E)$ by its covariance.** Let $\Sigma_E \triangleq \mathbb{E}[EE^\top]$. Among all distributions with covariance $\Sigma_E$, the Gaussian maximizes differential entropy, hence

$$h(E) \;\leq\; \frac{1}{2} \log\big((2\pi e)^d \det \Sigma_E\big). \tag{51}$$

**Step 2: relate $\det \Sigma_E$ to the total MSE.** By AM–GM on eigenvalues of $\Sigma_E$,

$$\det \Sigma_E \;\leq\; \Big(\frac{\mathrm{tr}(\Sigma_E)}{d}\Big)^d \;=\; \Big(\frac{\mathbb{E}\|E\|_2^2}{d}\Big)^d. \tag{52}$$

Plugging (52) into (51) yields

$$h(E) \;\leq\; \frac{d}{2} \log\Big(2\pi e \cdot \frac{\mathbb{E}\|E\|_2^2}{d}\Big). \tag{53}$$

**Step 3: lower bound $I(\theta; C)$ via conditional entropy.** We have

$$I(\theta; C) \;=\; h(\theta) - h(\theta \mid C). \tag{54}$$

Since $\widehat{\theta}(C)$ is measurable w.r.t. $C$, conditioning on $C$ also conditions on $\widehat{\theta}(C)$, hence

$$h(\theta \mid C) \;=\; h(\theta - \widehat{\theta}(C) \mid C) \;\leq\; h(\theta - \widehat{\theta}(C)) \;=\; h(E), \tag{55}$$

---

[2]This restriction matches the Bayes-optimal predictor in the linear-Gaussian model.

where the inequality uses that conditioning reduces entropy.

Combining (54) and (55) gives

$$I(\theta; C) \geq h(\theta) - h(E). \tag{56}$$

**Step 4: compute $h(\theta)$ and combine inequalities.** Since $\theta \sim \mathcal{N}(0, \tau^2 I_d)$,

$$h(\theta) = \frac{d}{2} \log(2\pi e \tau^2). \tag{57}$$

Substitute (53) and (57) into (56):

$$I(\theta; C) \geq \frac{d}{2} \log(2\pi e \tau^2) - \frac{d}{2} \log\left(2\pi e \cdot \frac{\mathbb{E}\|E\|_2^2}{d}\right) \tag{58}$$

$$= \frac{d}{2} \log\left(\frac{d\tau^2}{\mathbb{E}\|E\|_2^2}\right). \tag{59}$$

Exponentiating (59) and rearranging yields

$$\mathbb{E}\|E\|_2^2 \geq d\tau^2 \exp\left(-\frac{2}{d} I(\theta; C)\right), \tag{60}$$

which is (49). The expression (50) is the equivalent inversion of (60). $\qquad\square$

### B.1.3. PROOF OF THE LOWER BOUND IN THEOREM 4.1

*Proof of* (17)–(18). Fix any encoder $C = \text{Enc}(\mathcal{D}_N) \in \mathcal{V}^B$ and predictor $\widehat{y}(x_q, C)$. In the linear-Gaussian model, the Bayes-optimal predictor is linear in $x_q$, and by Lemma B.2 it suffices to lower bound $\mathbb{E}\|\theta - \widehat{\theta}(C)\|_2^2$. By Lemma B.3,

$$\mathbb{E}\left[\|\theta - \widehat{\theta}(C)\|_2^2\right] \geq d\tau^2 \exp\left(-\frac{2}{d} I(\theta; C)\right). \tag{61}$$

By Lemma B.1,

$$I(\theta; C) \leq \min\{B \log V, \; I(\theta; \mathcal{D}_N)\} = R_{\text{eff}}(B, N). \tag{62}$$

Since $u \mapsto \exp(-\frac{2}{d} u)$ is decreasing, combining (61) and (62) yields

$$\mathbb{E}\left[\|\theta - \widehat{\theta}(C)\|_2^2\right] \geq d\tau^2 \exp\left(-\frac{2}{d} R_{\text{eff}}(B, N)\right). \tag{63}$$

Finally, Lemma B.2 gives

$$\mathcal{R}(C) = \sigma^2 + \mathbb{E}\left[\|\theta - \widehat{\theta}(C)\|_2^2\right] \geq \sigma^2 + d\tau^2 \exp\left(-\frac{2}{d} R_{\text{eff}}(B, N)\right), \tag{64}$$

which is (17) with (18). $\qquad\square$

### B.1.4. MUTUAL INFORMATION OF SELECTED DEMONSTRATIONS

**Lemma B.4** (Closed form of $F(S)$). *Under* (14), *for any index set $S$ with $|S| = k$,*

$$F(S) = I(\theta; y_S \mid X_S) = \frac{1}{2} \log \det\left(I_d + \frac{\tau^2}{\sigma^2} X_S^\top X_S\right). \tag{65}$$

*Proof.* Condition on $X_S$. The vector $y_S \in \mathbb{R}^k$ satisfies

$$y_S = X_S \theta + \varepsilon_S, \qquad \varepsilon_S \sim \mathcal{N}(0, \sigma^2 I_k), \tag{66}$$

with $\theta \sim \mathcal{N}(0, \tau^2 I_d)$ independent of $\varepsilon_S$. Thus $y_S \mid X_S$ is Gaussian with covariance

$$\text{Cov}(y_S \mid X_S) = \tau^2 X_S X_S^\top + \sigma^2 I_k. \tag{67}$$

Using the mutual information identity for linear Gaussian models,

$$I(\theta; y_S \mid X_S) \; = \; h(y_S \mid X_S) - h(y_S \mid \theta, X_S). \tag{68}$$

We have $y_S \mid (\theta, X_S)$ is Gaussian with covariance $\sigma^2 I_k$, hence

$$h(y_S \mid \theta, X_S) \; = \; \frac{k}{2} \log(2\pi e \sigma^2). \tag{69}$$

Also, by (67),

$$h(y_S \mid X_S) \; = \; \frac{1}{2} \log\big((2\pi e)^k \det(\tau^2 X_S X_S^\top + \sigma^2 I_k)\big). \tag{70}$$

Subtracting (69) from (70) and simplifying gives

$$I(\theta; y_S \mid X_S) \; = \; \tfrac{1}{2} \log \det\Big(I_k + \tfrac{\tau^2}{\sigma^2} X_S X_S^\top\Big). \tag{71}$$

Finally, $\det(I_k + \alpha X_S X_S^\top) = \det(I_d + \alpha X_S^\top X_S)$ for any $\alpha > 0$, yielding

$$I(\theta; y_S \mid X_S) \; = \; \tfrac{1}{2} \log \det\Big(I_d + \tfrac{\tau^2}{\sigma^2} X_S^\top X_S\Big), \tag{72}$$

which is (65). $\qquad\square$

### B.1.5. SUBMODULARITY AND GREEDY APPROXIMATION

**Lemma B.5** (Monotonicity and submodularity of $F(S)$)**.** *Fix the covariates $\{x_i\}_{i=1}^N$ and define $F(S)$ as in (19). Then $F$ is monotone and submodular: for any $A \subseteq B \subseteq [N]$ and any $i \notin B$,*

$$F(A \cup \{i\}) - F(A) \; \geq \; F(B \cup \{i\}) - F(B). \tag{73}$$

*Proof.* Let $\alpha \triangleq \tau^2/\sigma^2$ and define $G(S) \triangleq I_d + \alpha X_S^\top X_S$. Then $F(S) = \frac{1}{2} \log \det(G(S))$.

**Step 1: express marginal gains.** For any set $S$ and element $i \notin S$, note that $X_{S \cup \{i\}}^\top X_{S \cup \{i\}} = X_S^\top X_S + x_i x_i^\top$. Hence

$$G(S \cup \{i\}) \; = \; G(S) + \alpha x_i x_i^\top. \tag{74}$$

By the matrix determinant lemma,

$$\det(G(S) + \alpha x_i x_i^\top) \; = \; \det(G(S)) \cdot \Big(1 + \alpha x_i^\top G(S)^{-1} x_i\Big). \tag{75}$$

Therefore the marginal gain is

$$F(S \cup \{i\}) - F(S) \; = \; \tfrac{1}{2} \log\Big(1 + \alpha x_i^\top G(S)^{-1} x_i\Big). \tag{76}$$

**Step 2: diminishing returns via Loewner order.** If $A \subseteq B$, then $X_A^\top X_A \preceq X_B^\top X_B$, so

$$G(A) \; \preceq \; G(B). \tag{77}$$

For positive definite matrices, Loewner order reverses under inversion, thus

$$G(A)^{-1} \; \succeq \; G(B)^{-1}. \tag{78}$$

Consequently,

$$x_i^\top G(A)^{-1} x_i \; \geq \; x_i^\top G(B)^{-1} x_i. \tag{79}$$

Since $u \mapsto \log(1 + \alpha u)$ is increasing for $u \geq 0$, applying it to (79) and using (76) yields (73). Monotonicity follows because the marginal gain (76) is nonnegative. $\qquad\square$

**Lemma B.6** (Greedy approximation for monotone submodular maximization)**.** *Let $F$ be monotone submodular and let $S^\star \in \arg\max_{|S|=k} F(S)$. Let $S_g$ be the size-$k$ greedy set that iteratively selects the element with the largest marginal gain. Then*

$$F(S_g) \geq \left(1 - \tfrac{1}{e}\right) F(S^\star). \tag{80}$$

*Proof.* Let $S_t$ be the greedy set after $t$ steps, with $S_0 = \emptyset$ and $|S_t| = t$. By submodularity and monotonicity,

$$F(S^\star) \leq F(S_t) + \sum_{i \in S^\star \setminus S_t} \big( F(S_t \cup \{i\}) - F(S_t) \big). \tag{81}$$

Since $|S^\star \setminus S_t| \leq k$, there exists at least one $i$ in $S^\star \setminus S_t$ with

$$F(S_t \cup \{i\}) - F(S_t) \geq \frac{1}{k} \big( F(S^\star) - F(S_t) \big). \tag{82}$$

Greedy picks an element with maximal marginal gain, so

$$F(S_{t+1}) - F(S_t) \geq \frac{1}{k} \big( F(S^\star) - F(S_t) \big). \tag{83}$$

Rearranging (83) gives the contraction

$$F(S^\star) - F(S_{t+1}) \leq \left(1 - \frac{1}{k}\right) \big( F(S^\star) - F(S_t) \big). \tag{84}$$

Iterating (84) for $t = 0, 1, \ldots, k-1$ yields

$$F(S^\star) - F(S_k) \leq \left(1 - \frac{1}{k}\right)^k F(S^\star) \leq e^{-1} F(S^\star), \tag{85}$$

hence $F(S_k) \geq (1 - 1/e) F(S^\star)$. Since $S_k = S_g$, this proves (80). $\qquad\square$

### B.1.6. BAYES RISK UPPER BOUND UNDER WELL-CONDITIONING

**Lemma B.7** (Bayes risk given $S$ and spectral bounds)**.** *Fix an index set $S$ with $|S| = k$, and consider the Bayes-optimal predictor given $(X_S, y_S)$. Let $\alpha \triangleq \tau^2/\sigma^2$ and define $G_S \triangleq X_S^\top X_S$. Then*

$$\mathcal{R}\big(C(S)\big) = \sigma^2 + \tau^2 \operatorname{tr}\Big(I_d + \alpha G_S\Big)^{-1}. \tag{86}$$

*Moreover, if $(1 - \delta)k I_d \preceq G_S \preceq (1 + \delta)k I_d$, then*

$$\mathcal{R}\big(C(S)\big) - \sigma^2 \leq \frac{d\tau^2}{1 + \alpha(1 - \delta)k}. \tag{87}$$

*Proof.* Condition on $(X_S, y_S)$. The posterior of $\theta$ is Gaussian with covariance

$$\Sigma_S = \Big(\tau^{-2} I_d + \sigma^{-2} G_S\Big)^{-1} = \tau^2 \Big(I_d + \alpha G_S\Big)^{-1}. \tag{88}$$

The Bayes-optimal predictor for $y_q$ is $\widehat{y} = x_q^\top \mathbb{E}[\theta \mid X_S, y_S]$. Using Lemma B.2 and the fact that the Bayes parameter MSE equals $\mathbb{E}[\|\theta - \mathbb{E}[\theta \mid X_S, y_S]\|_2^2] = \mathbb{E}[\operatorname{tr}(\Sigma_S)]$, we obtain

$$\mathcal{R}\big(C(S)\big) = \sigma^2 + \mathbb{E}\big[\operatorname{tr}(\Sigma_S)\big] = \sigma^2 + \tau^2 \operatorname{tr}\Big(I_d + \alpha G_S\Big)^{-1}, \tag{89}$$

which is (86).

For (87), the eigenvalues of $I_d + \alpha G_S$ lie in $[1 + \alpha(1 - \delta)k, \; 1 + \alpha(1 + \delta)k]$ under the spectral sandwich. Therefore every eigenvalue of $(I_d + \alpha G_S)^{-1}$ is at most $(1 + \alpha(1 - \delta)k)^{-1}$, so

$$\operatorname{tr}\Big(I_d + \alpha G_S\Big)^{-1} \leq \frac{d}{1 + \alpha(1 - \delta)k}. \tag{90}$$

Multiplying by $\tau^2$ and adding $\sigma^2$ proves (87). $\qquad\square$

**Lemma B.8** (Relating $F(S)$ to the RD exponent under (20)). *Assume $(1 - \delta)kI_d \preceq G_S \preceq (1 + \delta)kI_d$ and let $F(S)$ be defined in (19). Then*

$$\exp\left(-\tfrac{2}{d}F(S)\right) \in \left[\left(1 + \alpha(1 + \delta)k\right)^{-1}, \left(1 + \alpha(1 - \delta)k\right)^{-1}\right]. \tag{91}$$

*Consequently,*

$$\frac{1}{1 + \alpha(1 - \delta)k} \leq \kappa(\delta, k) \exp\left(-\tfrac{2}{d}F(S)\right), \tag{92}$$

*with $\kappa(\delta, k)$ as in (22).*

*Proof.* Under $(1 - \delta)kI_d \preceq G_S \preceq (1 + \delta)kI_d$, the eigenvalues $\{\lambda_j(G_S)\}_{j=1}^d$ satisfy $\lambda_j(G_S) \in [(1 - \delta)k, (1 + \delta)k]$. Hence

$$\det(I_d + \alpha G_S) = \prod_{j=1}^d \left(1 + \alpha\lambda_j(G_S)\right) \in \left[\left(1 + \alpha(1 - \delta)k\right)^d, \left(1 + \alpha(1 + \delta)k\right)^d\right]. \tag{93}$$

Since $F(S) = \tfrac{1}{2}\log\det(I_d + \alpha G_S)$, exponentiating $-\tfrac{2}{d}F(S) = -\tfrac{1}{d}\log\det(\cdot)$ gives

$$\exp\left(-\tfrac{2}{d}F(S)\right) = \det(I_d + \alpha G_S)^{-1/d}, \tag{94}$$

and combining (93) with (94) yields (91). For (92), use the lower end of (91):

$$\exp\left(-\tfrac{2}{d}F(S)\right) \geq \frac{1}{1 + \alpha(1 + \delta)k}. \tag{95}$$

Multiply both sides of (95) by $\kappa(\delta, k) = \frac{1 + \alpha(1 + \delta)k}{1 + \alpha(1 - \delta)k}$ to obtain

$$\kappa(\delta, k) \exp\left(-\tfrac{2}{d}F(S)\right) \geq \frac{1}{1 + \alpha(1 - \delta)k}, \tag{96}$$

which is (92). $\qquad\square$

### B.1.7. PROOF OF THE UPPER BOUND IN THEOREM 4.1

*Proof of (21)–(23).* By Lemma B.5, $F(S)$ in (19) is monotone submodular (conditioning on $\{x_i\}$). Thus Lemma B.6 applies and yields (23).

Next, apply Lemma B.7 with $S = S_g$ and the conditioning (20):

$$\mathcal{R}(C(S_g)) - \sigma^2 \leq \frac{d\tau^2}{1 + \alpha(1 - \delta)k}. \tag{97}$$

By Lemma B.8 with $S = S_g$,

$$\frac{1}{1 + \alpha(1 - \delta)k} \leq \kappa(\delta, k) \exp\left(-\tfrac{2}{d}F(S_g)\right). \tag{98}$$

Combining (97) and (98) gives

$$\mathcal{R}(C(S_g)) \leq \sigma^2 + \kappa(\delta, k)\, d\tau^2 \exp\left(-\tfrac{2}{d}F(S_g)\right). \tag{99}$$

Finally, using (23) and the monotonicity of $u \mapsto \exp(-\tfrac{2}{d}u)$,

$$\exp\left(-\tfrac{2}{d}F(S_g)\right) \leq \exp\left(-\tfrac{2}{d}\left(1 - \tfrac{1}{e}\right)F(S^\star)\right), \tag{100}$$

and substituting (100) into (99) yields (21). $\qquad\square$

### B.1.8. PROOF OF COROLLARY 4.2

*Proof.* Under (20), Lemma B.7 gives

$$\mathcal{R}(C(S_g)) - \sigma^2 \;\leq\; \frac{d\tau^2}{1 + \alpha(1 - \delta)k},\tag{101}$$

which is (24). To enforce $\mathcal{R}(C(S_g)) - \sigma^2 \leq \varepsilon$, it suffices that

$$\frac{d\tau^2}{1 + \alpha(1 - \delta)k} \;\leq\; \varepsilon.\tag{102}$$

Rearranging (102) and using $\alpha = \tau^2/\sigma^2$ yields

$$k \;\geq\; \frac{\sigma^2}{\tau^2} \cdot \frac{d}{(1 - \delta)\varepsilon},\tag{103}$$

which is (25). $\qquad\square$

## B.2. Full Proofs for Section 4.2

### B.2.1. PRELIMINARIES: (SUPER/SUB)MODULARITY AND MARGINALS

For a set function $H : 2^{\mathcal{N}} \to \mathbb{R}$ and sets $S \subseteq \mathcal{N}$, define the marginal gain

$$\Delta_H(i \mid S) \;\triangleq\; H(S \cup \{i\}) - H(S), \qquad \Delta_H(L \mid S) \;\triangleq\; H(S \cup L) - H(S),\tag{104}$$

for $i \in \mathcal{N} \setminus S$ and $L \subseteq \mathcal{N} \setminus S$.

A function $H$ is *submodular* if for all $A \subseteq B \subseteq \mathcal{N}$ and $i \in \mathcal{N} \setminus B$,

$$\Delta_H(i \mid A) \;\geq\; \Delta_H(i \mid B).\tag{105}$$

It is *supermodular* if the inequality is reversed.

### B.2.2. SUPERMODULARITY OF PAIRWISE SIMILARITY PENALTY

**Lemma B.9** (Pairwise similarity sum is supermodular). *Let $P(S) = \sum_{i<j,\; i,j\in S} s_{ij}$ with $s_{ij} = s_{ji} \geq 0$. Then $P$ is supermodular: for all $A \subseteq B \subseteq \mathcal{N}$ and $i \in \mathcal{N} \setminus B$,*

$$\Delta_P(i \mid A) \;\leq\; \Delta_P(i \mid B).\tag{106}$$

*Proof.* Fix any $S \subseteq \mathcal{N}$ and $i \notin S$. By expanding $P(S \cup \{i\})$,

$$\Delta_P(i \mid S) = P(S \cup \{i\}) - P(S)\tag{107}$$

$$= \sum_{j \in S} s_{ij}.\tag{108}$$

Now let $A \subseteq B$ and $i \notin B$. Since $A \subseteq B$ and $s_{ij} \geq 0$,

$$\sum_{j \in A} s_{ij} \;\leq\; \sum_{j \in B} s_{ij}.\tag{109}$$

Combining (108) and (109) yields

$$\Delta_P(i \mid A) \;\leq\; \Delta_P(i \mid B),\tag{110}$$

which is (106). $\qquad\square$

**Corollary B.10** ($-\lambda P$ is submodular). *For any $\lambda \geq 0$, the function $-\lambda P$ is submodular.*

*Proof.* By Lemma B.9, $P$ is supermodular, i.e., it satisfies (106). Multiplying (106) by $-\lambda \leq 0$ reverses the inequality:

$$-\lambda\,\Delta_P(i \mid A) \;\geq\; -\lambda\,\Delta_P(i \mid B),\tag{111}$$

which is exactly the diminishing-returns condition (105) for $-\lambda P$. $\qquad\square$

### B.2.3. SUBMODULARITY AND MONOTONICITY OF THE FULL OBJECTIVE

**Lemma B.11** (Modular functions are submodular). *Let $M(S) = \sum_{i \in S} m_i$ for fixed scalars $m_i$. Then $M$ is both submodular and supermodular.*

*Proof.* For any $A \subseteq B$ and $i \notin B$, the marginal gain is constant:

$$\Delta_M(i \mid A) \; = \; m_i \; = \; \Delta_M(i \mid B). \tag{112}$$

Thus (105) holds with equality, and the reverse inequality also holds. $\square$

**Lemma B.12** (Submodularity of $G$). *Assume $F$ is submodular. Define $G(S) = F(S) - \lambda P(S) - \alpha M(S)$ with $\lambda, \alpha \geq 0$. Then $G$ is submodular.*

*Proof.* By assumption, $F$ is submodular. By Corollary B.10, $-\lambda P$ is submodular. By Lemma B.11, $-\alpha M$ is submodular. Since a sum of submodular functions is submodular, we conclude that

$$G \; = \; F \; + \; (-\lambda P) \; + \; (-\alpha M) \tag{113}$$

is submodular. $\square$

**Proposition B.13** (Sufficient condition for monotonicity up to size $k$). *Assume $F$ is monotone. Let $\mu_k$ be defined in (29). If (30) holds, then for all $S \subseteq \mathcal{N}$ with $|S| \leq k - 1$ and all $i \notin S$,*

$$\Delta_G(i \mid S) \; \geq \; 0. \tag{114}$$

*Consequently, $G$ is monotone on all sets of size at most $k$.*

*Proof.* Fix any $S$ with $|S| \leq k - 1$ and any $i \notin S$. By expanding the marginal gain of $G$ and using (108) and modularity of $M$,

$$\Delta_G(i \mid S) = \Delta_F(i \mid S) \; - \; \lambda \, \Delta_P(i \mid S) \; - \; \alpha \, \Delta_M(i \mid S) \tag{115}$$

$$= \Delta_F(i \mid S) \; - \; \lambda \sum_{j \in S} s_{ij} \; - \; \alpha \, m_i. \tag{116}$$

Since $F$ is monotone submodular, $\Delta_F(i \mid S) \geq \mu_k$ by the definition (29). Also, $\sum_{j \in S} s_{ij} \leq |S| \, s_{\max} \leq (k-1) s_{\max}$ and $m_i \leq m_{\max}$, hence

$$\Delta_G(i \mid S) \; \geq \; \mu_k \; - \; \lambda(k-1) s_{\max} \; - \; \alpha m_{\max}. \tag{117}$$

Under (30), the right-hand side of (117) is nonnegative, proving (114). Monotonicity of $G$ on sets up to size $k$ follows because all single-element marginals are nonnegative along any chain of such sets. $\square$

### B.2.4. GREEDY APPROXIMATION FOR MONOTONE SUBMODULAR MAXIMIZATION

**Theorem B.14** (Greedy achieves $(1 - 1/e)$ for monotone submodular objectives). *Let $G$ be normalized, nonnegative, monotone, and submodular. Let $S^\star \in \arg\max_{|S| \leq k} G(S)$ and let $S_g$ be the size-$k$ greedy set defined by (28). Then*

$$G(S_g) \; \geq \; \left(1 - \tfrac{1}{e}\right) G(S^\star). \tag{118}$$

*Proof.* Let $S_t$ denote the greedy set after $t$ additions, so $S_0 = \emptyset$ and $|S_t| = t$. Define the residual optimality gap

$$\Delta_t \; \triangleq \; G(S^\star) - G(S_t). \tag{119}$$

By monotonicity, $G(S^\star \cup S_t) \geq G(S^\star)$, hence

$$G(S^\star \cup S_t) - G(S_t) \; \geq \; \Delta_t. \tag{120}$$

By submodularity (diminishing returns), the gain of adding the set $S^\star \setminus S_t$ is at most the sum of singletons evaluated at $S_t$:

$$G(S^\star \cup S_t) - G(S_t) \ \leq \ \sum_{i \in S^\star \setminus S_t} \Delta_G(i \mid S_t). \tag{121}$$

Combining (120) and (121) yields

$$\sum_{i \in S^\star \setminus S_t} \Delta_G(i \mid S_t) \ \geq \ \Delta_t. \tag{122}$$

Since $|S^\star \setminus S_t| \leq k$, there exists an element $i \in S^\star \setminus S_t$ with

$$\Delta_G(i \mid S_t) \ \geq \ \frac{1}{k}\Delta_t. \tag{123}$$

Greedy chooses an element with maximal marginal gain among all candidates, so in particular

$$G(S_{t+1}) - G(S_t) \ = \ \max_{i \in \mathcal{N} \setminus S_t} \Delta_G(i \mid S_t) \ \geq \ \frac{1}{k}\Delta_t. \tag{124}$$

Using (119) and (124), we obtain the contraction

$$\Delta_{t+1} = G(S^\star) - G(S_{t+1}) \tag{125}$$

$$\leq G(S^\star) - \left(G(S_t) + \tfrac{1}{k}\Delta_t\right) \tag{126}$$

$$= \left(1 - \tfrac{1}{k}\right)\Delta_t. \tag{127}$$

Iterating (127) from $t = 0$ to $t = k - 1$ yields

$$\Delta_k \ \leq \ \left(1 - \tfrac{1}{k}\right)^k \Delta_0 \ \leq \ e^{-1}\, G(S^\star), \tag{128}$$

where $\Delta_0 = G(S^\star) - G(\emptyset) = G(S^\star)$ since $G$ is normalized. Therefore,

$$G(S_g) \ = \ G(S_k) \ = \ G(S^\star) - \Delta_k \ \geq \ \left(1 - \tfrac{1}{e}\right) G(S^\star), \tag{129}$$

which is (118). $\qquad\square$

### B.2.5. WEAK SUBMODULARITY AND GREEDY APPROXIMATION

**Definition B.15** (Order-$k$ weak submodularity). *A nonnegative set function $G$ is* order-$k$ $\gamma_k$-weakly submodular *if there exists $\gamma_k \in (0, 1]$ such that for all $S \subseteq \mathcal{N}$ and all $L \subseteq \mathcal{N} \setminus S$ with $|L| \leq k$,*

$$\sum_{i \in L} \Delta_G(i \mid S) \ \geq \ \gamma_k\, \Delta_G(L \mid S). \tag{130}$$

**Theorem B.16** (Greedy achieves $(1 - e^{-\gamma_k})$ under weak submodularity). *Let $G$ be normalized, nonnegative, and monotone. Assume $G$ is order-$k$ $\gamma_k$-weakly submodular in the sense of (130). Let $S^\star \in \arg\max_{|S| \leq k} G(S)$ and let $S_g$ be the size-$k$ greedy set. Then*

$$G(S_g) \ \geq \ \left(1 - e^{-\gamma_k}\right) G(S^\star). \tag{131}$$

*Proof.* Let $S_t$ be the greedy set after $t$ steps and define $\Delta_t = G(S^\star) - G(S_t)$ as in (119). By monotonicity, $G(S^\star \cup S_t) \geq G(S^\star)$, hence

$$\Delta_G(S^\star \setminus S_t \mid S_t) \ = \ G(S^\star \cup S_t) - G(S_t) \ \geq \ \Delta_t. \tag{132}$$

Apply weak submodularity (130) with $S = S_t$ and $L = S^\star \setminus S_t$ (note $|L| \leq k$):

$$\sum_{i \in S^\star \setminus S_t} \Delta_G(i \mid S_t) \ \geq \ \gamma_k\, \Delta_G(S^\star \setminus S_t \mid S_t). \tag{133}$$

Combining (132) and (133) yields

$$\sum_{i \in S^\star \setminus S_t} \Delta_G(i \mid S_t) \ \geq \ \gamma_k \, \Delta_t. \tag{134}$$

Since $|S^\star \setminus S_t| \leq k$, there exists $i \in S^\star \setminus S_t$ with

$$\Delta_G(i \mid S_t) \ \geq \ \frac{\gamma_k}{k} \, \Delta_t. \tag{135}$$

Greedy chooses the maximum marginal, hence

$$G(S_{t+1}) - G(S_t) \ \geq \ \frac{\gamma_k}{k} \, \Delta_t. \tag{136}$$

Therefore the gap contracts as

$$\Delta_{t+1} = G(S^\star) - G(S_{t+1}) \tag{137}$$

$$\leq G(S^\star) - \left( G(S_t) + \tfrac{\gamma_k}{k} \Delta_t \right) \tag{138}$$

$$= \left( 1 - \tfrac{\gamma_k}{k} \right) \Delta_t. \tag{139}$$

Iterating (139) for $t = 0, 1, \dots, k-1$ yields

$$\Delta_k \ \leq \ \left( 1 - \tfrac{\gamma_k}{k} \right)^k \Delta_0 \ \leq \ e^{-\gamma_k} \, G(S^\star), \tag{140}$$

where $\Delta_0 = G(S^\star)$ by normalization. Thus,

$$G(S_g) \ = \ G(S_k) \ = \ G(S^\star) - \Delta_k \ \geq \ (1 - e^{-\gamma_k}) \, G(S^\star), \tag{141}$$

which is (131). □

## C. Additional Experimental Details

### C.1. Datasets and Splits

We evaluate on the union of dataset suites used in recent ICL selection / calibration / robustness papers:

- **Multi-task demo-selection benchmark (10 datasets)** from Liu et al. (2025): SST-5, MRPC, QNLI, CMSQA, HellaSwag (classification-style evaluation) and WebQuestions, GeoQuery, NL2Bash, MTOP, SMCalFlow (generation/semantic parsing). Following Liu et al. (2025), when an official test split is unavailable, we use the validation split for evaluation and the training split as the demonstration pool.

- **Dynamic calibration benchmark (8 datasets)** from Tan et al. (2025): SST-2, RTE, QNLI, MNLI, MRPC, WiC, YouTube-Spam, and AIGA. We follow their dynamic-ICL setting where each query has its own retrieved demonstrations.

- **OOD retrieval benchmark** from Wang et al. (2024) and used by Byun et al. (2026): QNLI, PIQA, WSC273, and Yelp as held-out (OOD) tasks, with a shared retrieval pool constructed from the remaining tasks in the benchmark. We additionally include MGSM as in Byun et al. (2026) for ID/OOD math evaluation.

- **Open-domain QA (retrieval augmentation)** from Kahardipraja et al. (2026): NQ-Swap and TriviaQA, evaluated under oracle / counterfactual / retrieved contexts.

- **Long-context + prompt-position sensitivity suite** from Cobbina & Zhou (2025): AG News, MNLI, ARC, MMLU, CNN/DailyMail, XSum, SQuAD, and GSM8K.

- **Intent classification + challenging reasoning** from Wang et al. (2025): BANKING77, CLINC150, HWU64, LIU54; and (for harder reasoning) BigBenchHard subsets (boolean expressions, web of lies), SST-5, and GSM8K.

## C.2. Metrics and Output Extraction

**Task metrics.** We use Acc. for classification; EM for structured generation/semantic parsing; Recall/EM for open-domain QA; ROUGE-L for summarization. For QA with potentially verbose outputs, we follow Kahardipraja et al. (2026) and treat Recall as primary.

**Token-efficiency metrics.** Let $B \in \mathcal{B}$ be a set of token budgets (e.g., $\{512, 1024, 2048, 4096, 8192\}$ depending on backbone limits). We report Acc@$B$ for each $B$, and Tok@Acc as

$$\text{Tok@Acc}(\tau) \triangleq \min\{B \in \mathcal{B} : \text{Acc}(B) \geq \tau\}, \tag{142}$$

for target accuracies $\tau$ chosen from baseline operating points. We additionally report an AUC over budgets when plotting scaling curves.

**Stability metrics.** Following Cobbina & Zhou (2025), we compute (i) Accuracy-Change: the absolute/relative accuracy difference induced by changing prompt placement and (ii) Prediction-Change: fraction of predictions flipped compared to the reference placement. In all main comparisons we fix placement to ssp (demos at the start), and use these metrics in controlled ablations.

**Calibration metrics.** We report NLL and ECE following calibration papers (Zhao et al., 2021; Zhou et al., 2024; Abbas et al., 2024; Tan et al., 2025). We compute ECE with 15 equal-width confidence bins unless otherwise noted.

## C.3. Baselines and Fair Comparison Protocol

**Training-free selection/retrieval.** We include Random; BM25 (Robertson et al., 2009); dense Top-$K$ using BERT/SBERT embeddings (Devlin et al., 2019; Reimers & Gurevych, 2019); and CEIL (Ye et al., 2023). All are run under the same candidate pool size $M$ and the same final token budget $B$.

**Learning-based selection / policy.** We include EPR (Rubin et al., 2022), LLM-R (Wang et al., 2024), RLDES (Wang et al., 2025), and CCL (Byun et al., 2026). These methods require extra training or learned policies/representations. To ensure comparability, we (i) fix the backbone LLM, (ii) fix the demonstration pool and retrieval candidate size $M$, and (iii) train any auxiliary selector/calibrator only on the *training* splits used by the original papers, never on test queries.

**Calibration baselines.** We include CC (Zhao et al., 2021), BC (Zhou et al., 2024), LinC (Abbas et al., 2024), and SC (Tan et al., 2025). These methods adjust predicted probabilities (or priors) given a context; they do not choose demonstrations. We therefore report them as composable layers on top of a shared demonstration selector (e.g., BM25), so that selection and calibration effects are disentangled.

## C.4. Prompting, Budgets, and Hyperparameters

**Prompt format.** We use a unified prompt template across all methods and place the demonstration block at the start of the system prompt (ssp) following Cobbina & Zhou (2025). All methods use identical label verbalizers and answer-format constraints.

**Budgets and candidate pools.** We set a fixed token budget $B$ and enforce $L(C) \leq B$ for every query. For each query, we retrieve $M$ candidates ($M$=50 by default) via BM25 or dense retrieval, then apply each selector to produce the final context.

**Hyperparameters.** We tune $(\lambda, \alpha)$ on a small grid $\lambda \in \{0, 0.05, 0.1, 0.2, 0.5\}$ and $\alpha \in \{0, 0.5, 1, 2\}$. Compaction tolerance uses $\varepsilon \in \{0, 10^{-3}, 10^{-2}, 5 \cdot 10^{-2}\}$. We select the best setting on the validation split of each suite and reuse it for that suite's test evaluation.

**Hardware and software.** We run inference on NVIDIA A100-class GPUs and implement all methods in PyTorch with HuggingFace Transformers; when available, we use vLLM for throughput. We fix random seeds for retrieval tie-breaking and for any stochastic components (e.g., Random selector).

## C.5. Experimental Setup for Greedy Selection Trajectory

For each query, we run the greedy selector (Eq. 11) under a fixed token budget $B$ and fixed candidate pool. At each selection step $t$, letting $S_t$ denote the selected set after step $t$, we log: $R_t \triangleq \sum_{\substack{i,j \in S_t \\ i<j}} s_{ij}$ (cumulative redundancy; Eq. 9),

$I_t \triangleq \sum_{i \in S_t} \rho_i(S_{<i})$ (cumulative information density; Eq. 8), $\quad K_t \triangleq \sum_{i \in S_t} c_i(S_{<i})$ (cumulative conflict; Eq. 10),

$g_t \triangleq \max_{i \notin S_t} \Delta(i \mid S_t)/\ell_i$ (current best marginal gain per token; Eq. 11). Here $S_{<i}$ denotes the prefix set right before selecting demonstration $i$. We stop when $g_t \leq 0$ or the budget is exhausted.

### C.6. Experimental Setup for KL-Bounded Compaction

We fix the candidate pool construction (same retriever, same pool size) and fix the selection penalties $(\lambda, \alpha)$ to the values chosen from the stability basin in Figure 2. For each budget $B \in \{64, 96, 128, 160, 192, 224, 256, 288, 320, 384, 448, 512\}$ and each divergence threshold $\varepsilon \in \{0.00, 0.04, \ldots, 1.20\}$, we run RDCO end-to-end (selection + compaction) and evaluate:

**(i) Score.** We report the **Overall** score (suite-averaged Acc./EM following Section 5.1).

**(ii) Drift.** For each query $x_q \in \mathcal{X}_{\text{probe}}$, let $C_{\text{pre}}$ be the selected (uncompacted) context and $C_{\text{post}}$ be the compacted context. Using the categorical distribution $q_\theta$ (Eq. 4), we measure prediction drift as

$$\text{Drift}(\varepsilon, B) \triangleq \frac{1}{|\mathcal{X}_{\text{probe}}|} \sum_{x_q \in \mathcal{X}_{\text{probe}}} D_{\text{KL}}\Big(q_\theta(\cdot \mid x_q, C_{\text{pre}}) \,\big\|\, q_\theta(\cdot \mid x_q, C_{\text{post}})\Big). \tag{143}$$

We visualize the resulting landscapes in Figure 4.

## D. Additional Experimental Results

### D.1. Efficiency Frontier: Performance–Cost Trade-off Under Inference Constraints

Although RDCO is training-free, it performs additional per-query computations. *does the inference overhead erase the practical benefit?* We therefore evaluate an efficiency frontier that jointly reports task performance and an inference-time cost proxy.

For a query $x_q$, method $m$ may invoke the backbone multiple times (e.g., candidate scoring + final decoding). Let $P_m(x_q)$ be the number of backbone forward passes used by method $m$, and let $L_u(x_q)$ be the prompt token length processed in pass $u \in \{1, \ldots, P_m(x_q)\}$. We define the **effective inference cost** as

$$\text{Cost}_m \triangleq \mathbb{E}_{x_q}\Big[ \sum_{u=1}^{P_m(x_q)} L_u(x_q) \Big], \tag{144}$$

reported in *kTok-equivalents* (thousands of tokens). This proxy captures both (i) prompt length and (ii) repeated model invocations, and can be replaced by measured wall-clock latency with the same analysis.

We sweep token budgets $B$ and candidate-pool sizes $N$ for each method, producing a dense set of cost–score points. For each configuration, we report the suite-averaged **Overall** score (matching Table 1) and the cost proxy in Eq. 144. We visualize the resulting cloud with a density contour (baselines) and explicitly plot the **Pareto frontier** for RDCO and representative baselines (Figure 1).

Figure 1 shows a clear separation between (i) extremely cheap but underperforming retrieval-only baselines, (ii) stronger but costlier learning-based selectors, and (iii) RDCO, which achieves a *better frontier* across a broad cost band. Notably, RDCO yields substantial gains in the low-to-mid cost regime where strict budgets make token allocation mistakes most damaging, while its advantage naturally saturates at high cost where many methods can afford large contexts. Overall, the frontier view indicates that RDCO's additional scoring/compaction overhead is *cost-effective* rather than prohibitive, strengthening its practical appeal under real inference constraints.

### D.2. Conflict Robustness Phase Diagram: Noise Rate $\times$ Conflict Penalty

We test whether increasing the conflict weight $\alpha$ systematically suppresses harmful demonstrations under controlled candidate-pool corruptions, and whether the resulting performance landscape exhibits a clear collapse boundary beyond which incorrect demonstrations dominate the prompt.

For each query, we build the same candidate pool as in the main setting (same retriever and pool size), then corrupt a fraction $r$ of demonstrations by randomly applying one of: (i) **label flip** (classification): replace $y_i$ with a random incorrect label; (ii) **wrong rationale** (generation): replace explanation/rationale spans with templated contradictory reasoning; (iii) **task**

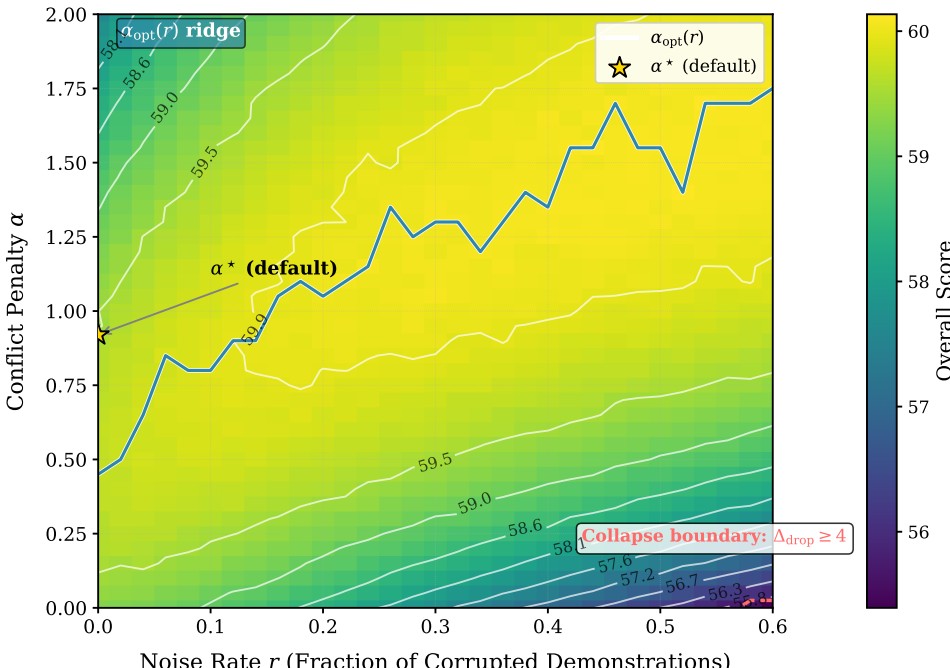

*Figure 5.* **Conflict robustness phase diagram over noise rate $r$ and conflict weight $\alpha$.** Heatmap shows the Overall score (higher is better). Solid contours are score level sets. The **dashed contour** marks the *collapse boundary* $\Delta_{\mathrm{drop}}(r, \alpha) \geq 4$ (Eq. 145), highlighting the region where corrupted demonstrations dominate and performance sharply degrades. The white curve traces $\alpha_{\mathrm{opt}}(r) = \arg\max_\alpha \mathrm{Score}(r, \alpha)$, showing that higher noise requires stronger conflict filtering, while overly large $\alpha$ becomes conservative and reduces score.

**mismatch** (format conflict): swap the output format to an incompatible schema (e.g., JSON vs. natural language). The corruption type is sampled uniformly from the three corruption categories.

We fix the token budget $B=256$ and redundancy weight $\lambda = \lambda^\star$ (from Figure 2). We sweep noise rate $r \in \{0.00, 0.02, \ldots, 0.60\}$ and conflict penalty $\alpha \in \{0.00, 0.05, \ldots, 2.00\}$. For each pair $(r, \alpha)$, we run RDCO end-to-end (selection + compaction) and report the suite-averaged **Overall** score. To visualize a performance "collapse" region, we define the drop relative to the clean best:

$$\Delta_{\mathrm{drop}}(r, \alpha) \triangleq \max_{\alpha'} \mathrm{Score}(0, \alpha') - \mathrm{Score}(r, \alpha), \tag{145}$$

and mark the *collapse boundary* where $\Delta_{\mathrm{drop}}(r, \alpha)$ exceeds a fixed threshold (e.g., 4 points).

Figure 5 reveals a clear *phase transition* consistent with the intended role of $\alpha$. At moderate-to-high noise, small $\alpha$ leads to a rapid score collapse, indicating that unfiltered conflicts can dominate the prompt and mis-specify the implicit task hypothesis. Increasing $\alpha$ shifts the system back into a stable, high-score region by downweighting inconsistent demonstrations. Importantly, the optimal ridge $\alpha_{\mathrm{opt}}(r)$ increases smoothly with $r$, demonstrating *controllability* rather than brittle tuning. Finally, very large $\alpha$ hurts even at high noise, suggesting over-regularization that discards informative (but slightly mismatched) evidence; this produces an interpretable trade-off boundary.

### D.3. Pool Sensitivity Heatmap: When Does RDCO Help Most?

Whether RDCO depends on a high-quality candidate pool, or whether it is most useful precisely when the pool is noisy, redundant, or conflict-prone? We therefore map RDCO's *relative gain* as a function of (i) the strict token budget $B$ and (ii) a *pool-quality* score computed from retrieval statistics. This yields a clear "applicability boundary": *poor pools* should benefit more from RDCO, while *very strong pools* should exhibit diminishing returns.

For each query, we form a candidate pool $\mathcal{P}(x_q)$ using the same retriever and pool size as in the main experiments. We

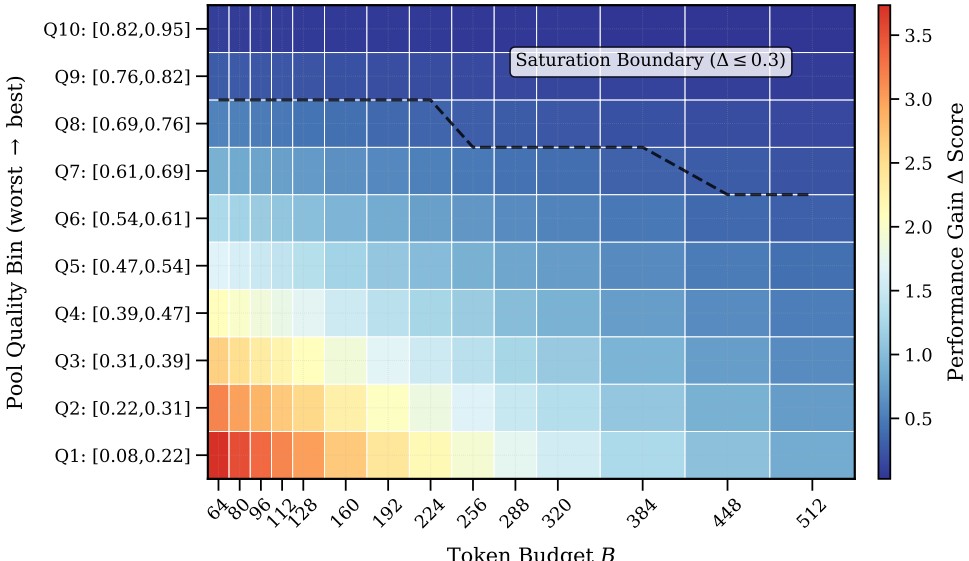

*Figure 6.* **Pool sensitivity heatmap over pool quality and token budget.** Color encodes RDCO's relative gain $\text{Gain}(B, \text{bin})$ (Eq. 147) against a fixed baseline under the same candidate pool and budget. The dashed contour marks a *saturation region* where gain is small (e.g., $\leq 0.3$ points), highlighting that RDCO is most beneficial when the candidate pool is poor and/or the budget is tight, while gains diminish for very high-quality pools and large budgets.

define a scalar pool-quality score $\text{Q}(x_q) \in [0, 1]$ by combining three normalized diagnostics:

$$\text{Q}(x_q) \triangleq 1 - \left( \omega_1 \widetilde{\text{Red}}(x_q) + \omega_2 \widetilde{\text{Conf}}(x_q) + \omega_3 \widetilde{\text{Flat}}(x_q) \right), \tag{146}$$

where $\widetilde{\text{Red}}$ is the mean pairwise similarity among retrieved demonstrations (higher means more redundancy), $\widetilde{\text{Conf}}$ is the fraction of candidates whose prefix-conditioned conflict score exceeds a threshold (Eq. 10), and $\widetilde{\text{Flat}}$ measures retrieval "flatness" via a normalized Top1–Top$k$ gap (e.g., smaller gaps imply less separation and thus lower quality). We use fixed weights $(\omega_1, \omega_2, \omega_3)$ and compute Q per query, then bucket queries into $M$ quantile bins from **worst** to **best** pool quality.

We sweep budgets $B$ (x-axis) and pool-quality bins (y-axis). For each $(B, \text{bin})$, we evaluate:

$$\text{Gain}(B, \text{bin}) \triangleq \text{Score}_{\text{RDCO}}(B, \text{bin}) - \text{Score}_{\text{BASE}}(B, \text{bin}), \tag{147}$$

where BASE is a fixed strong baseline under the *same* candidate pool and budget (e.g., Top-$K$ from the retriever with the same prompt template and compaction budget). We keep $(\lambda, \alpha, \varepsilon)$ fixed to their default values selected from Figures 2 and 4. Figure 6 visualizes $\text{Gain}(B, \text{bin})$ as a heatmap.

Figure 6 provides an actionable applicability boundary. When pool quality is low (high redundancy/conflict and weak separation), RDCO yields substantially larger gains, especially under tight budgets where token allocation mistakes are costly. As pool quality improves, the baseline already retrieves informative, non-conflicting demonstrations, and RDCO's marginal benefit shrinks, forming a clear saturation region. This pattern supports the interpretation that RDCO acts as a *pool denoiser and budget allocator*: it is a "firefighter" when retrieval is imperfect, and it gracefully degrades to near-neutral overhead when retrieval is already strong.

