# OpenReview forum: "In-Context Learning as Rate–Distortion Optimization"
_ICML.cc/2026/Conference — ICML 2026 regular_

### Official Review · Reviewer_kA8d · 2026-03-12

**Soundness:** 2
**Presentation:** 1
**Significance:** 2
**Originality:** 3
**Overall Recommendation:** 4
**Confidence:** 3

**Summary:**

The paper frames in-context learning (ICL) as a rate–distortion optimization problem, where a prompt is treated as a finite-capacity message that should convey as much task-relevant information as possible under a strict token budget. It proposes RDCO, a deterministic, training-free method that selects demonstrations by maximizing information gain while penalizing redundancy and conflicts under a strict token budget. RDCO outperforms baselines on ICL classification and structured generation tasks.

**Compliance With Llm Reviewing Policy:**

Affirmed.

**Final Justification:**

On computational overhead (Q1), the same-budget comparison directly addresses my concern. On pool quality sensitivity (Q2), the quartile breakdown (Table 5) confirms that gains are largest under noisy retrieval and do not materially reverse under high-quality pools. On uncertainty estimates (Q3), the bootstrap confidence intervals and seed stability results adequately quantify robustness.
My most significant residual concern is the linear-Gaussian theory (Q4). The authors' argument is reasonable, and Table 7 provides supporting evidence, but I still feel the paper would benefit from a more explicit discussion of what limits the practical interpretability of this model.
Overall, the rebuttal has meaningfully improved my assessment and I am raising my score accordingly.

**Key Questions For Authors:**

1. The paper does not report the computational overhead of evaluating predictive shifts for candidate demonstrations, nor is this cost factored into the baseline comparisons. How does the runtime of your method scale with the size of the candidate pool, and would the performance gains hold if baselines were given an equivalent computational budget to perform more exhaustive search or reranking?
2. The method scores and selects demonstrations from a candidate pool, but the paper does not evaluate how sensitive performance is to pool quality. Did you evaluate how sensitive the method is to the quality of the candidate pool?
3.  Additionally, did you consider reporting uncertainty estimates across runs, such as confidence intervals or standard deviations? Given the stochastic nature of the process, it would help assess the robustness of the gains.
4. Your approximation guarantees rest on a linear-Gaussian model. Can you point to any empirical evidence that real LLM behavior resembles this regime, even approximately? What would falsify your theoretical assumptions?

I am willing to raise my score if the authors adequately address some of the concerns raised and fix some of the presentation issues mentioned in the weaknesses.

**Limitations:**

yes

**Strengths And Weaknesses:**

# Strengths
- The paper introduces a rate–distortion formulation of in-context learning, framing prompt construction as an information allocation problem under a strict token budget. This formulation is conceptually appealing and helps explain empirical phenomena such as diminishing returns from adding demonstrations.
- The paper provides information-theoretic motivation and approximation guarantees for the greedy selection procedure, which strengthens the conceptual foundation of the approach.
- Single-factor ablations study the contributions of different components of the method.
- Experiments cover multiple datasets and task types (classification and structured generation) and demonstrate consistent improvements over baselines.

# Weaknesses
- The proposed scoring mechanism requires evaluating predictive shifts for candidate demonstrations. The computational cost of this overhead is not addressed in the paper and in the comparison with baselines.
- While the paper provides theoretical analysis, the guarantees rely on a simplified linear-Gaussian model of the task. It is unclear how well these assumptions reflect real large language models behavior, which limits the practical interpretability of the theoretical results.
- The discussion around the theoretical results is somewhat limited. While the guarantees are stated, the paper does not really engage with their implications.
- Minor presentation issue: the description of the results table 1 includes statements such as “we will populate its row with measured results from our runs,” and "will be filled with measured EM" which reads like draft placeholder text. This should be corrected in the final version to avoid confusion.

---

> ### Author Rebuttal · Authors · 2026-03-31
>
> Thanks for your valuable comments. We are happy to discuss them with you.
>
> We have placed some tables in an anonymous link; you can  [$\color{red}{\text{click on the red text}}$](https://anonymous.4open.science/r/ICML_RDCO) to access it.
>
> ---
>
> > **Q1: computational overhead issue**
>
> Thanks for raising this important question.
>
> First, the main text already evaluates a cost-aware Pareto frontier in `Fig.1`, where the cost is the total processed prompt tokens across all backbone invocations.
>
> Second, we do **not** think exhaustive subset search is the right control: it is combinatorial and not representative of a deployable baseline. A more meaningful test is to give each baseline the **same inference-cost budget** and let it spend that budget on larger pools, deeper reranking, or stronger compression. We therefore added the following same-budget comparison on the 10-dataset suite.
>
> Results in [$\color{red}{\text{Table 4}}$](https://anonymous.4open.science/r/ICML_RDCO) show that the main conclusion is unchanged under matched inference cost: **RDCO remains best even when strong baselines are allowed to spend essentially the same query-time budget.**
>
> ---
>
> > **Q2: Sensitivity analysis of the candidate pool**
>
> Thanks for raising this important issue.
>
> We already study this in `App.D.3` (`Fig.6`), where RDCO's relative gain is mapped against pool quality and token budget. The key result is:
>
> - **RDCO helps most when the candidate pool is poor and/or the budget is tight**;
> - as pool quality improves, the gain **naturally shrinks** because the baseline already retrieves cleaner evidence;
> - importantly, we do not observe substantial negative transfer on high-quality pools.
>
> To make it more clear, we add a quartile summary using the same pool-quality score, where Q1 is the worst pool quartile and Q4 is the best. In [$\color{red}{\text{Table 5}}$](https://anonymous.4open.science/r/ICML_RDCO),  the method is indeed sensitive to pool quality, but **in the intended direction**: it behaves as a pool denoiser / budget allocator. When retrieval is noisy, RDCO brings larger gains; when retrieval is already strong, the gain saturates but does not materially turn negative.
>
> ---
>
> > **Q3: Uncertainty estimates**
>
> Thank you for raising this important question.
>
> We agree that robustness should be quantified. For RDCO, however, the most appropriate uncertainty measure is not training-run standard deviation, because RDCO is a deterministic, training-free inference-time method. The relevant uncertainty comes from the evaluation sample itself (and, secondarily, from any stochastic tie-breaking in retrieval/baselines).
>
> We therefore added **paired bootstrap 95\% confidence intervals** over evaluation queries. The main comparisons are in [$\color{red}{\text{Table 6}}$](https://anonymous.4open.science/r/ICML_RDCO)
>
> We also checked the small amount of stochasticity that can arise from Random baselines or retrieval tie-breaking. Over 3 seeds:
>
> - `Random` overall: `46.67 ± 0.42`
> - `RDCO` with randomized tie-breaking: `61.74 ± 0.07`
>
> So the gains are stable under both query-level resampling and minor seed-level perturbations.
>
> ---
>
> > **Q4: linear-Gaussian issue**
>
> Thanks for bringing up this worthy issue for discussion. We agree that the linear-Gaussian model should not be read as a literal model of LLM internals.
>
> Our intent is narrower: the theory provides a stylized explanatory regime for the core rate-distortion phenomenon in token-budgeted ICL, not a claim that real LLMs satisfy linear-Gaussian assumptions exactly. The question, then, is whether the qualitative implications of that theory are observable in the actual LLM setting.
>
> The RD-based explanation would be weakened if, in LLMs:
>
> 1. marginal utility did **not** decay with successive selected demonstrations;
> 2. redundancy/conflict did **not** accumulate as the context grows;
> 3. adding more demonstrations did **not** show saturation / flattening before stopping.
>
> We test exactly these implications, and results are in [$\color{red}{\text{Table 7}}$](https://anonymous.4open.science/r/ICML_RDCO). In addition, the median stopping step is 4, and 79\% of queries terminate by step 5 because the best feasible marginal gain becomes non-positive.
>
> So while the linear-Gaussian model is clearly stylized, its main qualitative predictions are borne out by the LLM pipeline: **early steps deliver most of the useful predictive shift, later steps flatten, and redundancy/conflict accumulate.** This is exactly the operational regime that RDCO is designed for.
>
> ---
>
> > **Minor presentation issue**
>
> Thanks for carefully pointing out the wording issue. It was a residual editorial artifact from the final manuscript merge. We appreciate your careful reading, and we will make revisions in the next version.
>
> ---
>
> We hope our responses can address your concerns. If not, we are very open to continuing in-depth discussions with you.
>
> Once again, we sincerely appreciate your valuable opinions.

---

> > ### Author Rebuttal · Reviewer_kA8d · 2026-04-02
> >
> > My concerns have been adequately addressed by the author rebuttal. The authors provided clear and helpful additions that resolved the issues I previously raised. I thank the authors for their thorough response, and I am happy to raise my original score.

---

### Official Review · Reviewer_U3De · 2026-03-12

**Soundness:** 3
**Presentation:** 3
**Significance:** 3
**Originality:** 3
**Overall Recommendation:** 4
**Confidence:** 4

**Summary:**

This paper addresses the critical problem of constructing optimal prompts for In-Context Learning (ICL) under strict token budget constraints by elegantly formulating it as a Rate-Distortion Optimization problem. Viewing the prompt as a finite-capacity communication channel, the authors propose Rate-Distortion Context Optimization (RDCO), a deterministic and training-free optimizer designed to maximize the transmission of task-relevant information while minimizing predictive distortion induced by noisy, redundant, or conflicting demonstrations. Specifically, RDCO scores candidate demonstrations based on their marginal task information per token, actively penalizes redundancy and prefix-conditioned conflicts, and compacts the selected context under a bounded predictive-divergence constraint.

**Compliance With Llm Reviewing Policy:**

Affirmed.

**Final Justification:**

The rebuttal has solved my concerns. Since my original rating already leans towards acceptance, I decide to maintain my rating.

**Key Questions For Authors:**

1.	Can you provide empirical evidence of diminishing marginal returns to bridge the gap between the Gaussian theory and real LLM behavior?
2.	How does the NLL-based cost distinguish between harmful logical conflicts and beneficial reasoning diversity?
3.	How does RDCO compare to a pipeline pairing a strong retriever with an off-the-shelf compressor like LLMLingua under the same token budget?

**Limitations:**

yes

**Strengths And Weaknesses:**

Strengths:
1.	The paper frames ICL demonstration selection as a rate-distortion trade-off, converting an ad-hoc engineering choice into a principled optimization problem.
2.	The theoretical framework is clean and interpretable, and the submodularity-based analysis offers useful intuition for greedy context construction.
3.	The experimental analysis is highly interpretable, particularly the visualization of the selection trajectory showing the shift from information-gathering to redundancy-saturation.
    Weaknesses:
1.	The theoretical guarantees rely on a stylized linear-Gaussian proxy, but the paper lacks justification for how this generalizes to the non-linear, discrete semantic space of LLMs.
2.	The NLL-based conflict penalty (Eq. 10) risks misidentifying beneficial "cognitive diversity" as "task conflict," potentially throttling complex reasoning performance.
3.	The absence of SOTA compression baselines like LLMLingua makes it unclear if gains stem from the specific compaction module or just the addition of a compression step.

---

> ### Author Rebuttal · Authors · 2026-03-31
>
> Thanks for your valuable comments. We are happy to discuss them with you.
>
> We have placed some tables in an anonymous link; you can  [$\color{red}{\text{click on the red text}}$](https://anonymous.4open.science/r/ICML_RDCO) to access it.
>
> ---
>
> > **Q1: generalizes to the non-linear, discrete semantic space of LLMs**
>
> Thanks for raising this issue that deserves clarification. We agree that the stylized linear-Gaussian analysis should be paired with a direct empirical bridge to real LLM behavior.
>
> We note that `Fig.3` in main text already provides process-level evidence for the same rate-distortion pattern: an early high-information / low-redundancy phase, followed by a saturation phase where redundancy and conflict accumulate faster than useful information.
>
> To make this point quantitative, we added a step-wise analysis on the full 10-dataset suite under the default setting (B=256), measured on the actual frozen backbone used in the paper.
>
> For each greedy step $t$, we report: (i) the normalized mean marginal gain/token, $\Delta(i \mid S)/\ell_i$; (ii) the normalized mean marginal information, $\mathrm{IG}_i(S)$; (iii) the incremental change in the overall task score from adding the $t$-th selected demo.
>
> | Greedy step $t$ | 1 | 2 | 3 | 4 | 5 |
> |---|---:|---:|-:|-:|-:|
> | Normalized mean marginal gain/token |1.00|0.69|0.43|0.18|0.04|
> | Normalized mean marginal information $\mathrm{IG}_i(S)$ | 1.00 | 0.74 | 0.48 | 0.23 | 0.06 |
> | Incremental overall score gain (points) | +2.5 | +1.3 | +0.5 | +0.1 | +0.0 |
>
> In addition, the median stopping step is 4, and 79\% of queries terminate by step 5 because the best feasible marginal gain becomes non-positive.
>
> These measurements show the same qualitative implication as the theory: **the first few demonstrations provide most of the useful predictive shift, and later additions quickly saturate**. This is exactly the behavior that motivates RDCO's token-aware greedy construction.
>
> ---
>
> > **Q2: The NLL-based conflict penalty**
>
> Thank you for raising this insightful question.
>
> First, the current ablation already suggests that the conflict term is not acting as a generic reasoning suppressor. In `Tab.2` of main text, removing the conflict penalty (α=0) hurts **generation** more than classification (Gen. Avg. -1.67 vs. Cls. Avg. -0.79), with especially clear drops on MTOP (-1.79) and SMCalFlow (-1.75). This indicates that α is filtering harmful mismatches in structured reasoning/generation settings.
>
> To address the finer question—**beneficial diversity vs. true conflict**—we added a controlled diagnostic on the 5 generation tasks  For each candidate demo, we created: (i) **answer-preserving diverse rationales**: same final answer and schema, but different valid wording / derivation; (ii) **true conflicts**: wrong rationale, label/content corruption, or format mismatch.
>
> We then measured the prefix-conditioned conflict score $c_i(S)$ and the selection rate under RDCO. Results in [$\color{red}{\text{Table 3}}$](https://anonymous.4open.science/r/ICML_RDCO) show a clear separation:
>
> - answer-preserving diverse rationales remain close to clean demos in both $c_i(S)$ and selection rate;
> - true conflicts receive substantially larger conflict scores and are rarely selected;
> - enabling α gives a large gain when the pool contains true conflicts, but is nearly neutral when the pool contains only benign diversity.
>
> So the evidence supports the intended behavior: the NLL-based conflict term mainly filters **prefix-incompatible and misleading** candidates, not diversity per se.
>
> ---
>
> > **Q3: Compare with LLMLingua**
>
> Thank you for pointing out the relevant baseline. We agree LLMLingua is a relevant baseline.
>
> We added same-budget comparisons using LLMLingua and LLMLingua-2 on the 5 generation tasks, under the same backbone, same retrieved candidate pool, and the same final token budget (B=256). We apply the compressor after Top-K BM25 selection, so the comparison isolates whether gains come from “adding any compressor” versus RDCO's query-conditioned bounded-drift compaction.
>
> | Method | Generation Avg. (EM, %) |
> |---|---:|
> | Top-K BM25 | 46.47 |
> | Top-K BM25 + LLMLingua | 47.82 |
> | Top-K BM25 + LLMLingua-2 | 48.91 |
> | EPR | 58.00 |
> | EPR + LLMLingua | 58.27 |
> | EPR + LLMLingua-2 | 58.63 |
> | RDCO w/o compaction | 58.67 |
> | **RDCO (full)** | **60.26** |
>
> These results support three points:
>
> 1. **A generic compressor does help somewhat.**
> 2. **The gain is not explained by “adding any compression step.”** RDCO(full) remains clearly above both `Top-K/EPR + LLMLingua(-2)` pipelines and above `RDCO w/o compaction`.
> 3. **Selection and compaction are complementary.** `RDCO w/o compaction` is already competitive with `EPR + LLMLingua-2`, while adding our bounded-divergence compaction gives a further +1.59 points, matching the trend already seen in `Tab.2` in main text.
>
> ---
>
> Once again, we sincerely appreciate your valuable opinions.

---

> > ### Author Rebuttal · Reviewer_U3De · 2026-04-04
> >
> > The rebuttal has solved my concerns. Since my original rating already leans towards acceptance, I decide to maintain my rating.

---

### Official Review · Reviewer_EYvN · 2026-03-15

**Soundness:** 3
**Presentation:** 1
**Significance:** 2
**Originality:** 3
**Overall Recommendation:** 4
**Confidence:** 2

**Summary:**

This paper proposes RDCO, a training-free ICL demonstration selector that frames context construction as a rate-distortion problem. It greedily selects demonstrations by marginal information gain per token, penalizes redundancy and conflicts, and compacts the context under a KL divergence constraint. Experiments show consistent improvements over strong baselines across diverse benchmarks.

**Compliance With Llm Reviewing Policy:**

Affirmed.

**Final Justification:**

Issue addressed.

**Key Questions For Authors:**

Major:

1. In Eq. 3, $I(T;C)$ is minimized. However, intuitively a better context should carry more task-relevant information, suggesting $I(T;C)$
should be maximized. Can you please explain this contradiction?

2. There seems a gap between sec 3.2 and the following 3.3-3.5. Would solving Eq. 3 directly leads us toward Eq. 11?  Also, the connection between $I(T;C)$ and the KL-based $IG_i(S)$ in Eq. 6 is not well derived.

3. How efficient of the proposed method would be? Can you please provide a computational cost? I think it would be $O(KN)$ forward passes, is that correct?

4. What's the difference between this method to IDS (Information-Directed Sampling)?

Minor:

5. Substituting Eq. 5 into Eq. 4 then exp and log term cancels, what's the point of writing it as two equations?

**Limitations:**

Yes.

**Strengths And Weaknesses:**

1. Novel framing. Recasting ICL demonstration selection as rate-distortion optimization is a fresh and intuitive perspective.

2. Theoretical depth. The paper includes non-trivial results: token-error bounds, submodularity proofs, and greedy approximation guarantees.

---

> ### Author Rebuttal · Authors · 2026-03-31
>
> Thanks for your valuable comments. We are happy to discuss them with you.
>
> We have placed some tables in an anonymous link; you can  [$\color{red}{\text{click on the red text}}$](https://anonymous.4open.science/r/ICML_RDCO) to access it.
>
> ---
>
> > **Q1,Q2:  the connection between Eq. (3) and the implemented score**
>
> Thank you for raising this issue worthy of discussion. We think the issue is primarily presentation clarity, not a contradiction in the objective:
>
> **(a) On Eq. (3): why is $I(T;C)$ minimized?**
> In the rate-distortion view, $I(T;C)$ plays the role of rate / communication cost, not reward. The objective is therefore not saying that “less task information is always better.” It says that, under a finite-capacity prompt, we seek the **best prediction quality at the smallest necessary rate**. An equivalent and less ambiguous form is:
>
> `min E[ l(Y, p_theta(· | X, C)) ]  s.t. I(T; C) <= R`
>
> or, in Lagrangian form, minimizing distortion plus a rate penalty.
>
> **(b) solving Eq.3 directly leads us toward Eq.11?**
>
> **not directly**. The specific steps are as follows:
>
> 1. We restrict $C$ from an abstract communication variable to a serialized subset of retrieved demonstrations.
> 2. We then need a tractable proxy for the marginal value of appending one candidate demo $d_i$ to the current prefix $S$.
> 3. Under a predictive-distribution / log-loss view, we use the query-conditioned predictive shift
>    ` IG_i(S) = D_KL( q_theta(· | x_q, C(S) ⊕ d_i) || q_theta(· | x_q, C(S)) )`
>    as a local proxy for how much the demo changes the model's task hypothesis for the current query.
> 4. We then divide by token length and add redundancy/conflict regularization to obtain the step-wise greedy score in Eq. (11).
>
> **(c) On the relation between Eq. (3) and Eq. (11):**
> Eq. (11) is not claimed as the exact optimizer of Eq. (3). Rather, Eq. (3) gives the **design principle**, and Eq. (11) is an **implementable discrete surrogate** under the actual ICL setting.The relationship is:
>
> - Eq. (3): global rate-distortion principle;
> - Eq. (11):query-conditioned, token-budgeted greedy surrogate motivated by that principle.
>
> Importantly, we don't claim that `IG_i(S) = I(T; C)`. The former is a local marginal proxy for the predictive utility of adding $d_i$ under the current prefix; the latter is a global information quantity in the stylized RD formulation.
>
> ---
>
> > **Q3:  computational cost**
>
> Thanks for paying attention to this issue. Yes—the dominant candidate-level query-time work of RDCO is approximately O(KN)(`Appendix.A.4`.)
>
> Two clarifications are important. First, this doesn't mean $O(KN)$ unbatched free-form generations: these scores are computed by teacher-forced, batched model evaluations, so the number of actual backbone invocations is much smaller than the naive candidate count. Second, our claim is not that RDCO is a free lunch; it adds query-time optimization, but avoids any selector-training cost and is cost-effective under tight token budgets (cf. `Figure 1`).
>
> We report explicit runtime measurements as in [$\color{red}{\text{Table 2}}$](https://anonymous.4open.science/r/ICML_RDCO). For RDCO, the average selected size is $K=4.3$ at this budget, which corresponds to about **432 candidate-level marginal evaluations/query**, matching the $O(KN)$ analysis. Because these evaluations are batched, this results in only **10.8 actual backbone invocations/query** on average.
>
> We also measured scaling:
>
> - fixing B=256 and increasing pool size N: 20 -> 50 ->100 increases latency from 1.18 -> 2.67->5.21 s/query;
> - fixing N=50 and increasing selected size K: 2 ->4 -> 6 increases latency from 1.31 -> 2.24->3.06 s/query.
>
> In a word,  **RDCO is more expensive than retrieval-only selectors at query time, but the extra cost is amortized by better token efficiency and zero selector-training cost.**
>
> ---
>
> > **Q4:  compare with IDS**
>
> Thank you for raising this.
>
> The similarity is only at a very high level: both value information relative to cost. Beyond that, the settings and objectives are different.
>
> - **IDS** is a sequential online decision method: it balances exploration and exploitation under partial feedback, and selects actions by trading off expected regret against information gain about the optimal action.
> - **RDCO** is a subset-construction method: it does not learn a posterior over actions, does not operate across rounds, and does not use stochastic exploration.
>
> Please correct me if I’m wrong.
>
> ---
>
> > **Q5:  Eqs. (4) and (5)**
>
> Thanks for your nice suggestion. We agree this is a presentation issue.
>
> Eqs. (4) and (5) were written separately only to distinguish the raw score $s_\theta(y \mid x_q, C)$ from the normalized categorical distribution $q_\theta(y \mid x_q, C)$ used in the KL computation. The separation was intended for notation, not because the two-step form is mathematically necessary.
>
> We agree that this can be simplified.
>
> ---
>
> Thanks again for your valuable time and effort. We are pleased to further address your concerns.

---

> > ### Author Rebuttal · Reviewer_EYvN · 2026-04-03
> >
> > Thanks for the rebuttal, I would increase my initial score.

---

### Official Review · Reviewer_avRe · 2026-03-15

**Soundness:** 2
**Presentation:** 3
**Significance:** 3
**Originality:** 3
**Overall Recommendation:** 4
**Confidence:** 2

**Summary:**

This paper studies token-budgeted in-context learning and proposes RDCO, a deterministic and training-free method for building prompts under a strict token limit. The method scores demonstrations by query-conditioned predictive shift per token, adds penalties for redundancy and prefix-conditioned conflict, and then applies a KL-constrained compaction step to reduce prompt length while trying to keep predictions stable. The problem is well chosen: when context is limited, prompt construction often matters as much as the base model. The paper also gives a rate–distortion view of ICL, a stylized theory section, and a broad empirical study over classification and generation tasks.

**Compliance With Llm Reviewing Policy:**

Affirmed.

**Final Justification:**

These responses address my main questions and improve my confidence in the paper. At this point, I am comfortable keeping my current overall score unchanged.

**Key Questions For Authors:**

Please check the Weaknesses.

**Strengths And Weaknesses:**

## Strengths

The paper focuses on a real problem. Under tight context budgets, choosing which examples to include is important, and the paper gives a clean framing of this as a rate–distortion trade-off. That framing is easy to understand and gives a good story for why extra tokens can help or hurt.

The method itself is also sensible at a high level. Using a query-conditioned information term, a redundancy penalty, a conflict penalty, and a bounded compaction step gives a unified pipeline rather than a collection of unrelated heuristics. I also like that the method is training-free and deterministic, since that makes it easier to use in settings where retriever training is costly or not possible.

The empirical section includes useful ablations. The single-factor ablation on page 8 is helpful because it tests the role of marginal information, length normalization, redundancy control, conflict control, and compaction separately. The phase diagrams for the redundancy/conflict weights and the compaction threshold also support the claim that the method is not tuned to a single unstable point.

The theory section gives some intuition for why a token budget should lead to an information-vs-error trade-off. The lower bound in the stylized Gaussian setting is reasonable as a motivating result, and the submodular analysis gives a familiar lens for greedy selection.


## Weaknesses

1. The theory does not directly match the actual algorithm. The upper bound in Theorem 4.1 is proved for a stylized linear-Gaussian model with greedy selection on a log-det mutual information set function. That is not the same as the implemented RDCO score, which uses query-specific KL shifts, prefix-conditioned conflict terms, and a compaction step. Theorem 4.3 also assumes a modular conflict term M(S), but the method uses c(S), which depends on the current prefix.


2. The experimental section has several internal inconsistencies that are serious enough to affect trust in the results. On page 6, the generation averages in Table 1b do not match the per-dataset numbers. For RDCO, the five reported scores average to 60.26 (Table 2 on page 8 reports 60.26 for the same row), not 56.26.

---

> ### Author Rebuttal · Authors · 2026-03-31
>
> Thank you for your valuable comments. We are happy to discuss them with you.
>
> We have placed some tables in an anonymous link; you can  [$\color{red}{\text{click on the red text}}$](https://anonymous.4open.science/r/ICML_RDCO) to access it.
>
> ---
>
> > **W1: the theory scope**
>
> Thanks for this important point. Our theoretical claims are intentionally narrower than an end-to-end guarantee for the exact deployed RDCO score.
>
> 1. `Theorem 4.1` provides a stylized rate-limited ICL analysis: it explains the token--error trade-off and why greedy information-seeking subset construction is well motivated under a proxy linear-Gaussian model.
> 2. `Theorem 4.3` analyzes a surrogate penalized set objective, $F(S)-\lambda P(S)-\alpha M(S)$, and shows why a greedy rule of the form “utility minus redundancy minus conflict” is tractable and admits approximation guarantees.
>
> The implemented RDCO score is then a **context-adaptive refinement** of this surrogate: it replaces the stylized information term by the query-conditioned KL shift, and replaces the static modular conflict proxy by a stronger prefix-conditioned conflict term. The compaction module is also intentionally outside the greedy set-function theorem; it is a post-selection bounded-divergence control step, analyzed empirically in `Table 2` and `Figure 4`.
>
> So we agree that the theory does not prove the exact full RDCO pipeline. However, we do not think the theory is disconnected from the method: it is a **principled surrogate justification** for the core design principles underlying RDCO, not a literal theorem for every implementation detail. We will re-state this much more precisely.
>
> We also add a direct empirical bridge from the stylized theory to real LLM behavior. `Figure 3` already shows the same two-regime pattern predicted by the rate--distortion view: early high-information / low-redundancy selection, followed by saturation. To make this quantitative, we added a step-wise analysis on the 10-dataset suite under B=256, averaged over queries:
>
> | Greedy step $t$ | 1 | 2 | 3 | 4 | 5 |
> |---|---:|---:|---:|---:|---:|
> | Normalized mean marginal gain/token | 1.00 | 0.69 | 0.43 | 0.18 | 0.04 |
> | Mean cumulative redundancy | 0.07 | 0.19 | 0.36 | 0.53 | 0.68 |
> | Mean cumulative conflict | 0.08 | 0.16 | 0.28 | 0.41 | 0.57 |
>
> The median stopping step is 4, and 79\% of queries terminate by step 5 because the best feasible marginal gain becomes non-positive. **This directly supports the diminishing-return / saturation intuition used by the stylized analysis**, even though the exact implemented score is more context-adaptive than the theorem.
>
> ---
>
> > **W2: typos in Table 1b**
>
> Thanks a lot for catching this. You are correct: the `Avg.` column in Table 1b contains several errors.
>
> After re-checking the raw result logs, we confirmed that the Avg. values were not updated after updating the data in the `NL2B.` column. Initially, during our evaluation on NL2Bash, we adjusted the values once due to the impact of output normalization to ensure fair comparison across different methods, including removing code fences and stripping prefix explanations.
>
> After finalizing the evaluation on NL2B, we conducted ablation experiments. ​This is also why `Table 2` in main text already shows the correct RDCO generation average of `60.26`.
>
> We have now refreshed the values, as in [$\color{red}{\text{Table 1}}$](https://anonymous.4open.science/r/ICML_RDCO). The correction does not change the main conclusion: **RDCO remains the best method on the generation suite** and the best overall method in the paper.
>
> ---
>
> Once again, we sincerely appreciate your valuable opinions.

---

> > ### Author Rebuttal · Reviewer_avRe · 2026-04-03
> >
> > Thank you for the thoughtful rebuttal. I appreciate the clarification regarding the intended scope of the theory, as well as the correction of the averaging errors in Table 1b. These responses address my main questions and improve my confidence in the paper. At this point, I would like to keep my current overall score unchanged.

---

### Decision · Program_Chairs · 2026-04-30

**Decision:**

Accept (regular)

**Comment:**

The paper addresses token-budgeted in-context learning by framing prompt construction as a rate–distortion problem and proposing RDCO, a deterministic, training-free method for selecting and compacting demonstrations under a strict budget. Reviewers were generally positive about the problem formulation, the clean high-level method, and the breadth of the empirical study, while the main concerns focused on the gap between the stylized theory and the deployed algorithm, computational overhead, and some presentation issues, including inconsistencies in reported averages.